# Reassessment of long-period constituents for tidal predictions along the German North Sea coast and its tidally influenced rivers

Andreas Boesch[1] and Sylvin Müller-Navarra[1]

[1]Bundesamt für Seeschifffahrt und Hydrographie, Bernhard-Nocht-Straße 78, 20359 Hamburg, Germany

**Correspondence:** Andreas Boesch (andreas.boesch@bsh.de)

**Abstract.** The Harmonic Representation of Inequalities is a procedure for tidal analysis and prediction that combines aspects of the non-harmonic and the harmonic method. With this technique, the deviations of heights and lunitidal intervals, especially of high and low waters, from their respective mean values are represented by superpositions of long-period tidal constituents. This article documents the preparation of a constituents list for the operational application of the Harmonic Representation of Inequalities. Frequency analyses of observed heights and lunitidal intervals of high and low water from 111 tide gauges along the German North Sea coast and its tidally influenced rivers have been carried out using the generalized Lomb-Scargle periodogram. One comprehensive list of partial tides is realized by combining the separate frequency analyses and by applying subsequent improvements, e.g. through manual inspections of long time series data. The new set of 39 partial tides largely confirms the previously used set with 43 partial tides. Nine constituents are added and 13 partial tides, mostly in close neighbourhood of strong spectral components, are removed. The effect of these changes has been studied by comparing predictions with observations from 98 tide gauges. Using the new set of constituents, the standard deviations of the residuals are reduced on average by 2.41% (times) and 2.30% (heights) for the year 2016. The new set of constituents will be used for tidal analyses and predictions starting with the German tide tables for the year 2020.

## 1 Introduction

Tidal predictions for the German Bight are calculated at the Federal Maritime and Hydrographic Agency (*Bundesamt für Seeschifffahrt und Hydrographie*, BSH) and are published in official tide tables each year. The preparation of tidal predictions has a long tradition at BSH and its predecessor institutions: the first tide tables by the German Imperial Admiralty were issued for the year 1879.

Since 1954, a method named Harmonic Representation of Inequalities (HRoI) has been used at BSH to calculate tidal predictions for tide gauge locations along the German North Sea coast and its tidally influenced rivers (Horn, 1948, 1960; Müller-Navarra, 2013). This technique allows analysing the deviations of times and heights, especially at high and low water, from their respective mean values. In contrast to the widely used harmonic method (e.g. Parker, 2007, and references therein), the HRoI utilizes only long-period partial tides. This reduction in frequency space allows for a computationally efficient way to calculate times and heights of high and low water. Other techniques for tidal analysis of high and low waters are described, e.g., in Doodson (1951) and Foreman and Henry (1979); these two methods additionally consider diurnal and semi-

diurnal constituents. The HRoI has proven to be especially useful for predicting semi-diurnal tides in shallow waters where the harmonic method would need a large number ($\gtrsim 60$) of constituents or could even fail to produce adequate results. The fundamentals of the HRoI are summarized in Sect. 2 for completeness.

An important aspect of tidal prediction is the selection of relevant partial tides (angular velocities, $\omega$) to be included in the underlying analysis of water level records. While it is possible to determine these partial tides individually for each single tidal analysis, it is desirable in an operational service to have one comprehensive set of constituents that can be used for all tide gauges under investigation. Horn (1960) presented a list of 44 angular velocities that were used with the HRoI. This selection of partial tides was probably utilized until the year 1969 when the set was slightly modified (compare Tab. 2 in Sect. 2). To our knowledge, no documentation exists of the methods and specific water level records that were used to prepare these lists of angular velocities.

The objective of this work is to review the set of partial tides used with the HRoI by determining the most important long-period constituents for applications in the German Bight. Therefore, we perform a spectral analysis of water level observations from 111 tide gauges. The available tide gauge data is presented in Sect. 3. The analysis of high and low water time series is described in Sect. 4. In Sect. 5, tidal predictions based on an existing list of partial tides and predictions based on the new set are compared with observed water levels. The article closes with a comparison of predictions made with the HRoI and the harmonic method for two sites (Sect. 6) and the conclusions (Sect. 7).

## 2  Harmonic Representation of Inequalities

The Harmonic Representation of Inequalities (HRoI) is a derivative of the non-harmonic method by essentially translating it into an analytical form. The non-harmonic method has been used for a long time, e.g. by Lubbock (1831) for the analysis of tides in the port of London. With the non-harmonic method, the times of high and low waters are calculated by adding mean lunitidal intervals and corresponding inequalities to the times of lunar transits. Likewise, the heights of high and low waters are determined by adding corresponding inequalities to the respective mean heights. The inequalities are corrections for the relative positions of earth, moon and sun (e.g. semi-monthly, parallactic, declination).

The original implementation of the HRoI, as introduced by Horn (1948, 1960), can be used to calculate vertices of tide curves, i.e. high water time, high water height, low water time and low water height. In this form the method is tailored to semi-diurnal tides. Müller-Navarra (2013) shows how the HRoI may be generalized to predict tidal heights at equidistant fractions of the mean lunar day. This generalization allows the determination of the full tidal curve at a chosen sampling interval. Here, we focus only on the application of calculating the times and heights of high and low waters.

According to Horn (1960), the HRoI combines the best from the harmonic and the non-harmonic method: the analytical procedure of the first method, and the principle of calculating isolated values directly which is characteristic for the second. The strength of the HRoI lies in the prediction of times and heights of high and low water when the full tidal curve is considerably non-sinusoidal. This is frequently the case in shallower waters, such as the German Bight, and in rivers. As the HRoI uses only

**Table 1.** The high and low waters are classified into four types (event index $k$).

| $k$ | description |
| --- | --- |
| 1 | high water assigned to upper transit |
| 2 | low water assigned to upper transit |
| 3 | high water assigned to lower transit |
| 4 | low water assigned to lower transit |

observed times and/or heights of high and/or low waters, the method can also be applied when a record of the full tidal curve is not available (e.g. historic data) or when a tide gauge runs dry around low water (e.g. analysis of only high waters).

Let $(t_j, h_j), j = 1, ..., J$, be a time series of length $J$ of high and low water heights $h_j$ recorded at times $t_j$. All times need to be given in UTC. The HRoI method is based on the assumption that the variations of the individual heights and lunitidal

intervals around their respective mean values can be described by sums of harmonic functions. The lunitidal interval is the time difference between the time $t_j$ and the corresponding lunar transit at Greenwich. As a general rule, the daily higher high water and the following low water are assigned to the previous upper lunar transit, and the daily lower high water and the following low water are assigned to the previous lower transit. For example, in the year 2018, the mean lunitidal interval for high (low) water was determined to be 9 h 4 min (16 h 5 min) for Borkum and 15 h 22 min (22 h 32 min) for Hamburg. See Fig. 1 in Sect. 3

for the locations of these two sites.

A convenient method to organize high and low waters of semi-diurnal tides is the lunar transit number $n_t$ (Müller-Navarra, 2009). It counts the number of upper lunar transits (unit symbol: tn) at the Greenwich meridian since the transit on December 31, 1949, which has been arbitrarily set to $n_t = 0$ tn. A lower transit always has the same transit number as the preceding upper transit. Each high and low water is uniquely identified by using the number $n_t$ of the assigned lunar transit and an additional

event index $k$ as defined in Tab. 1. The differentiation between upper and lower transit allows for changes in the Moon's declination which alternately advance and retard times, and increase and decrease the heights of successive tides (diurnal inequality).

A full tidal analysis with the HRoI comprises the investigation of eight time series (heights and lunitidal intervals of the four event types listed in Tab. 1). Each time series is described by a model function $\hat{y}$ of the following form:

$$\hat{y}(n_t) = a_0 + \sum_{l=1}^{L} \left[ a_l \cos(\omega_l n_t) + a_{l+L} \sin(\omega_l n_t) \right]. \tag{1}$$

The parameters $a_l, l = 0, ..., 2L$ are determined from a least-squares fit, i.e.

$$\chi^2 = \sum_{j=0}^{J} (y_j - \hat{y}_j)^2 \rightarrow \min \quad , \tag{2}$$

where $y_j$ are the observed heights or lunitidal intervals. The angular velocities $\omega_l$ [°/tn] are taken from a previously defined set of $L$ partial tides. In Tab. 2, we list two sets of partial tides that have been used in the past at BSH and the new set that is

the result of this work.

**Table 2.** Sets of angular velocities that have been used with the HRoI. See Sect. 2 for a description of the columns.

| Doodson | $m_s$ | $m_h$ | $m_p$ | $m_{N'}$ | $\omega$ [°/h] | $\omega$ [°/tn] | name | set 1[a] | set 2[b] | this work |
|---------|-------|-------|-------|----------|----------------|-----------------|------|----------|----------|-----------|
| ZZZZAZ | 0 | 0 | 0 | 1 | 0.0022064 | 0.0548098 | | x | x | x |
| ZZZAZZ | 0 | 0 | 1 | 0 | 0.0046418 | 0.1153082 | | | x | |
| ZZZBZZ | 0 | 0 | 2 | 0 | 0.0092836 | 0.2306165 | | | | x |
| ZZAYZZ | 0 | 1 | -1 | 0 | 0.0364268 | 0.9048862 | | | x | |
| ZZAZZZ | 0 | 1 | 0 | 0 | 0.0410686 | 1.0201944 | Sa | x | x | x |
| ZZBXZZ | 0 | 2 | -2 | 0 | 0.0728537 | 1.8097724 | | x | x | x |
| ZZBZZZ | 0 | 2 | 0 | 0 | 0.0821373 | 2.0403886 | Ssa | x | x | x |
| ZAXZZZ | 1 | -2 | 0 | 0 | 0.4668792 | 11.5978420 | | x | x | x |
| ZAXAZZ | 1 | -2 | 1 | 0 | 0.4715211 | 11.7131503 | MSm | x | x | x |
| ZAYXZZ | 1 | -1 | -2 | 0 | 0.4986643 | 12.3874200 | | | | x |
| ZAYZZZ | 1 | -1 | 0 | 0 | 0.5079479 | 12.6180365 | | | | x |
| ZAYAAZ | 1 | -1 | 1 | 1 | 0.5147961 | 12.7881545 | | | | x |
| ZAZYYZ | 1 | 0 | -1 | -1 | 0.5421683 | 13.4681129 | | x | x | |
| ZAZYZZ | 1 | 0 | -1 | 0 | 0.5443747 | 13.5229227 | Mm | x | x | x |
| ZAZZYZ | 1 | 0 | 0 | -1 | 0.5468101 | 13.5834211 | | x | x | |
| ZAZZZZ | 1 | 0 | 0 | 0 | 0.5490165 | 13.6382309 | | x | x | x |
| ZAZZAZ | 1 | 0 | 0 | 1 | 0.5512229 | 13.6930407 | | x | x | x |
| ZAZAZZ | 1 | 0 | 1 | 0 | 0.5536583 | 13.7535391 | | | | x |
| ZABYZZ | 1 | 2 | -1 | 0 | 0.6265120 | 15.5633115 | | x | x | |
| ZABBAZ | 1 | 2 | 2 | 1 | 0.6426438 | 15.9640460 | | | | x |
| ZBVBZZ | 2 | -4 | 2 | 0 | 0.9430421 | 23.4263005 | | x | | |
| ZBWZZZ | 2 | -3 | 0 | 0 | 0.9748271 | 24.2158785 | | x | x | x |
| ZBXZYZ | 2 | -2 | 0 | -1 | 1.0136894 | 25.1812631 | | x | x | x |
| ZBXZZZ | 2 | -2 | 0 | 0 | 1.0158958 | 25.2360729 | MSf | x | x | x |
| ZBXZAZ | 2 | -2 | 0 | 1 | 1.0181022 | 25.2908827 | | x | x | |
| ZBXAZZ | 2 | -2 | 1 | 0 | 1.0205376 | 25.3513811 | | x | x | |
| ZBYZZZ | 2 | -1 | 0 | 0 | 1.0569644 | 26.2562673 | | | | x |
| ZBZXZZ | 2 | 0 | -2 | 0 | 1.0887494 | 27.0458453 | | x | x | x |
| ZBZYZZ | 2 | 0 | -1 | 0 | 1.0933912 | 27.1611535 | | x | x | x |
| ZBZZYZ | 2 | 0 | 0 | -1 | 1.0958266 | 27.2216520 | | x | x | |
| ZBZZZZ | 2 | 0 | 0 | 0 | 1.0980330 | 27.2764618 | Mf | x | x | x |

| Doodson | $m_s$ | $m_h$ | $m_p$ | $m_{N'}$ | $\omega$ [°/h] | $\omega$ [°/tn] | name | set 1[a] | set 2[b] | this work |
|---------|-------|-------|-------|----------|---------------|----------------|------|----------|----------|-----------|
| ZBZZAZ | 2 | 0 | 0 | 1 | 1.1002394 | 27.3312716 | | x | x | x |
| ZCVAZZ | 3 | -4 | 1 | 0 | 1.4874168 | 36.9492232 | S$\nu$2 | x | x | x |
| ZCWYZZ | 3 | -3 | -1 | 0 | 1.5192018 | 37.7388011 | | x | x | |
| ZCXYYZ | 3 | -2 | -1 | -1 | 1.5580641 | 38.7041858 | | x | x | |
| ZCXYZZ | 3 | -2 | -1 | 0 | 1.5602705 | 38.7589956 | SN | x | x | x |
| ZCXYAZ | 3 | -2 | -1 | 1 | 1.5624769 | 38.8138054 | | x | x | |
| ZCXZZZ | 3 | -2 | 0 | 0 | 1.5649123 | 38.8743038 | | x | x | x |
| ZCXAZZ | 3 | -2 | 1 | 0 | 1.5695541 | 38.9896120 | MStm | x | x | x |
| ZCZWZZ | 3 | 0 | -3 | 0 | 1.6331241 | 40.5687675 | | x | | |
| ZCZYZZ | 3 | 0 | -1 | 0 | 1.6424077 | 40.7993844 | Mfm | x | x | x |
| ZDUZZZ | 4 | -5 | 0 | 0 | 1.9907229 | 49.4519514 | | x | x | x |
| ZDVZZZ | 4 | -4 | 0 | 0 | 2.0317915 | 50.4721458 | 2SM | x | x | x |
| ZDXXZZ | 4 | -2 | -2 | 0 | 2.1046452 | 52.2819182 | | x | x | |
| ZDXZZZ | 4 | -2 | 0 | 0 | 2.1139288 | 52.5125347 | MSqm | x | x | x |
| ZDXZAZ | 4 | -2 | 0 | 1 | 2.1161352 | 52.5673444 | | | | x |
| ZDZZZZ | 4 | 0 | 0 | 0 | 2.1960661 | 54.5529235 | | x | x | x |
| ZETAZZ | 5 | -6 | 1 | 0 | 2.5033126 | 62.1852961 | | x | x | x |
| ZEVYZZ | 5 | -4 | -1 | 0 | 2.5761662 | 63.9950685 | 2SMN | x | x | x |
| ZEVZZZ | 5 | -4 | 0 | 0 | 2.5808080 | 64.1103767 | | x | | |
| ZEVAZZ | 5 | -1 | 1 | 0 | 2.5854499 | 64.2256849 | | | | x |
| ZEXYZZ | 5 | -2 | -1 | 0 | 2.6583035 | 66.0354573 | | x | x | |
| ZFTZZZ | 6 | -6 | 0 | 0 | 3.0476873 | 75.7082187 | | x | x | x |
| ZFVZZZ | 6 | -4 | 0 | 0 | 3.1298246 | 77.7486076 | | x | x | x |
| ZHRZZZ | 8 | -8 | 0 | 0 | 4.0635830 | 100.9442917 | | x | x | x |
| Number of partial tides in set of constituents: | | | | | | | | 44 | 43 | 39 |

[a] set 1 was probably used until the year 1969, see also Tab. 3 in Horn (1960). [b] set 2 was probably used from 1970 until 2019, see also appendix E in Goffinet (2000), Tab. 5 in Müller-Navarra (2013) includes $\omega = 23.4263005°$/tn but this angular velocity has never been included in calculations for BSH tide tables or tide calendars.

All tidal constituents considered here have angular velocities that are linear combinations of the rate of change of four fundamental astronomical arguments: the mean longitude of the moon ($s$), the mean longitude of the sun ($h$), the mean longitude of the lunar perigee ($p$) and the negative of the longitude of the moon's ascending node ($N'$). The second to fifth column in Tab. 2 give the respective linear coefficients $m$. The two other arguments that one encounters using the harmonic method can be effectively neglected: the coefficients for the rate of change of the mean lunar time and of the mean longitude of the solar perigee are always equal to zero, because only long-period constituents need to be considered, and the time series are too short to resolve differences due to the variations of the solar perigee. The angular velocities in the sixth and seventh column are given

in degrees per hour and in degrees per transit number, respectively. The conversion between these two units is $1°/\text{tn} \cdot \tau[\frac{\text{h}}{\text{tn}}] = 1°/\text{h}$ with the length of the mean lunar day $\tau = 24.84120312 \frac{\text{h}}{\text{tn}}$. The angular velocities are calculated using the expressions for the fundamental astronomical arguments as published by the International Earth Rotation and Reference Systems Service (2010, Sect. 5.7). The alphabetical Doodson number is given in the first column (Doodson, 1921; Simon, 2013). The eighth column

states the commonly used names[1]. A mark in one of the last three columns indicates whether the angular velocity is included in the respective constituents list for usage with the HRoI.

## 3   Tide gauge data

The tide gauges at the German coast and in rivers are operated by different federal and state authorities. These agencies provide BSH with quality-checked water level records of high and low waters (times and heights). Table A1 in the appendix lists 137

German tide gauges which deliver water level observations on a regular basis and for which tidal predictions were published in BSH tide tables (*Gezeitentafeln*) or tide calendar (*Gezeitenkalender*) for the year 2018 (Bundesamt für Seeschifffahrt und Hydrographie, 2017a, b). For the analysis presented in Sect. 4, all data until the year 2015 is considered that was systematically archived in electronic form at the BSH tidal information service (as of August 2018). The data periods are given in the fourth and fifth column in Tab. A1 and cover $22 - 27$ years for most gauges. Much longer time series were readily available for tide

gauges at Cuxhaven (BSH gauge number 506P) and Hamburg (508P) for which data since the year 1901 is used. We are aware that the tidal regime can change over such a long time, but include all available data in the analysis to maximize the achievable spectral resolution.

     Only tide gauges with more than 19 years of data are included in order to cover the period of rotation of the lunar node (18.6 years) in the frequency analysis. In addition, we use only tide gauges where more than 60% of high and low waters are

recorded during the gauge's data period. This criterion excludes gauges for which no low water observations are available. The 111 gauges that fulfil these two criteria are marked in the column labelled "used for analysis" in Tab. A1. The locations of all tide gauges are shown on the map in Fig. 1.

## 4   Analysis of high water and low water time series

The following analysis is applied to the water level records of all 111 tide gauges that are marked in the seventh column of

Tab. A1 in the appendix.

### 4.1   Data preparation

Data preparation includes the assignment of lunar transit numbers $n_t$ and the calculation of lunitidal intervals as described in Sect. 2 for each record of high or low water. The lunar transit times are calculated following the algorithm by Meeus

---

[1]see, e.g., the IHO Standard List of Tidal Constituents: https://www.iho.int/mtg_docs/com_wg/IHOTC/IHOTC_Misc/TWCWG_Constituent_list.pdf

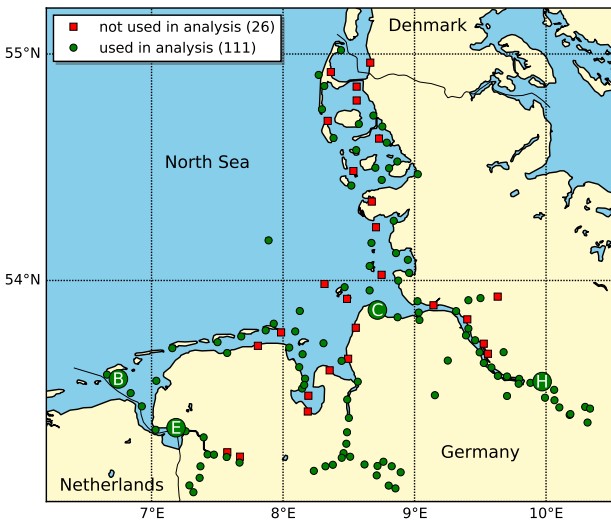

**Figure 1.** The locations of all tide gauges in the German Bight from Tab. A1. Some of the tide gauges mentioned in the text are highlighted: Borkum, Fischerbalje (B); Emden, Große Seeschleuse (E); Cuxhaven, Steubenhöft (C); Hamburg, St. Pauli (H).

(1998, Chap. 15) with the modification of direct calculation of lunar coordinates using the periodic terms given in the work by Chapront-Touzé and Chapront (1991).

The observed water levels include extreme events, such as storm surges. These events are not representative for the tidal behaviour at the site of a tide gauge and are removed from the data set. We apply a 3-sigma-clipping separately for the eight times series analysed with the HRoI (see Sect. 2). Only those data points are used in the analysis, for which the height and the lunitidal interval are within the range of three times the respective standard deviation.

## 4.2 Frequency analysis

The observed heights and lunitidal intervals ($y$) can be understood as being functions of the assigned transit number ($n_t$). We calculate periodograms for the heights and tidal intervals using the corresponding frequency scale tn$^{-1}$.

The occurrences of high and low waters are irregularly spaced in time. Additionally, there are many longer data gaps which cannot be interpolated. This excludes the fast Fourier transform (FFT) as a spectral analysis technique. Instead, we use the generalized Lomb-Scargle periodogram as defined by Zechmeister and Kürster (2009), including their normalization if not mentioned otherwise. The frequency scale covers the range from 0.0001 to 2 tn$^{-1}$ with an interval of 0.01999 tn$^{-1}$ (100 000 points in the periodogram). This corresponds to approximately $0.0057 - 114.5916°$/tn or $0.0002 - 4.6130°$/h. The upper limit corresponds to twice the mean sampling interval (Nyquist criterion).

Artefacts from spectral leakage pose a major problem when identifying peaks in a periodogram. They arise from the finite length of the time series. This effect can be reduced by applying an apodization function, i.e. multiplying the data with a suitable window function, that smoothly brings the recorded values to zero at the beginning and the end of the sampled time

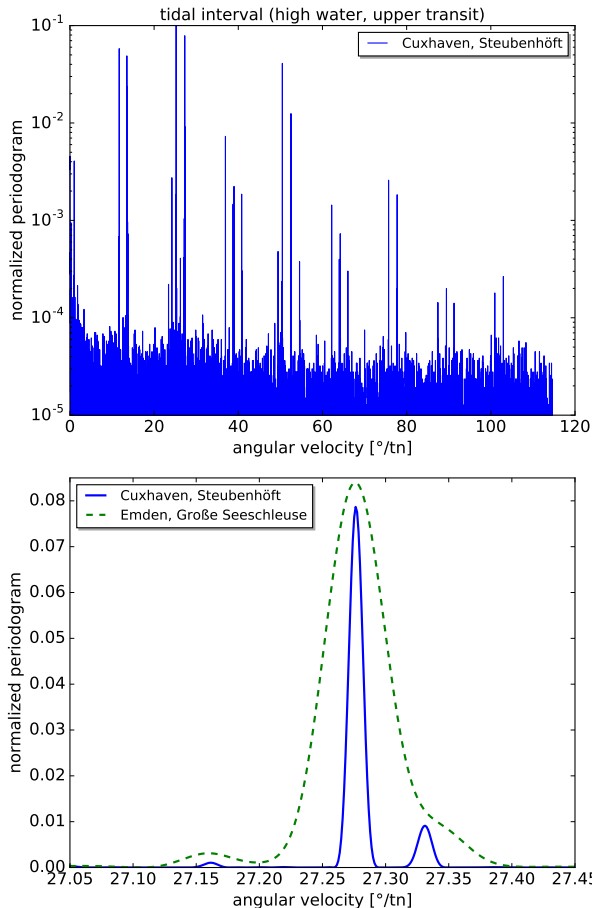

**Figure 2.** *Upper panel:* Normalized periodogram of the lunitidal intervals of high waters (assigned to upper lunar transits) for the tide gauge Cuxhaven. Notice the upper part of the logarithmic scale is truncated at 0.1 for better visibility of weak lines. *Lower panel:* Zoom into the region with the spectral line corresponding to half a tropical month (Mf) at 27.2764618°/tn. The longer time series for Cuxhaven leads to narrower spectral lines (solid blue curve) compared to Emden (dashed green line).

series (e.g. Press et al., 1992; Prabhu, 2014). We apply a Hanning window to the data which gives a good compromise between reducing side lobes and preserving the spectral resolution.

For each tide gauge, periodograms are calculated for the eight time series that are analysed with the HRoI. In the upper panels of Fig. 2 and 3, we show periodograms of the lunitidal intervals and heights (of high waters assigned to an upper transit, event index $k = 1$) for the tide gauge Cuxhaven. Cuxhaven (together with Hamburg) provides by far the longest time series that is used in the analysis (compare Tab. A1). In these figures, the vertical axis is normalized to the strongest peak and the horizontal axis is converted to degrees per transit number for better comparison with Tab. 2. The periodogram for the lunitidal intervals reveals many more strong spectral lines above the noise floor as compared to the periodogram for the heights. A

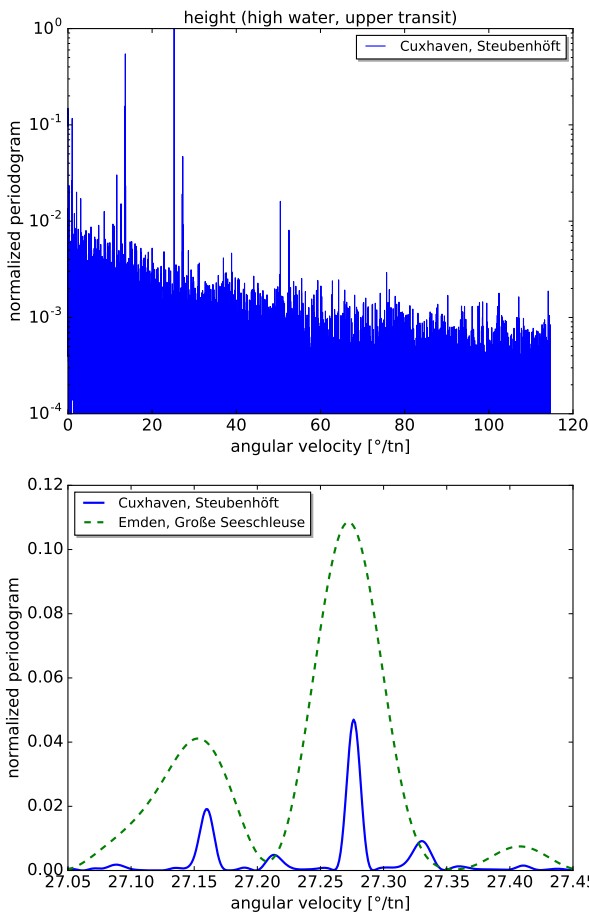

**Figure 3.** *Upper panel:* Normalized periodogram of the heights of high waters (assigned to upper lunar transits) for the tide gauge Cuxhaven. Notice the logarithmic scale. *Lower panel:* Zoom into the region with the spectral line corresponding to half a tropical month (Mf) at 27.2764618°/tn. The longer time series for Cuxhaven leads to narrower spectral lines (solid blue curve) compared to Emden (dashed green line).

frequency-dependent noise level can clearly be seen in Fig. 3 (noise level increases towards lower angular velocities). The lower panels of Fig. 2 and 3 show a small extract of the respective upper periodograms. Additionally, data for tide gauge Emden is included for illustration of the differences in spectral line width. The time series from Emden is about four times shorter than the one from Cuxhaven. This leads to broader spectral lines in the periodogram and it can be expected that some
5   weaker lines are unresolvable.

## 4.3 Identifying relevant partial tides

We aim to find all local maxima in a periodogram that are above a noise threshold. This threshold is calculated in a two-step process that is described in the following.

In the first step, the strongest spectral lines are removed from the periodogram. The values above the 99.5th percentile are removed from the data set and a histogram is calculated from the remaining values $p$ (100 bins with central values $x_{\mathrm{bin}}$). The histogram shows an exponential trend from a large number of data points with low periodogram values to a few points that fall into the bins at the upper end. An exponential curve $y_{\mathrm{bin}} = a \cdot \exp(-x_{\mathrm{bin}}/b)$ is fitted to the histogram, with fit parameters $a$ and $b$. The process of removing data points above the 99.5th percentile from the periodogram is repeated until the ratio $\max(p)/b$ falls below the value of 30. This value is based on experience.

In the second step, the noise threshold is determined using a set of remaining points in the periodogram that represent a continuum above the noise level. The result is illustrated in Figs. 4 and 5 for lunitidal intervals and heights at the tide gauge Borkum. For this procedure, the periodogram is split into 25 sections with the same number of data points. The data point at the 99.5th percentile is selected in each section and an exponential function is fitted to these 25 points. The fit is repeated after a 1-sigma-clipping. The noise threshold corresponds to the resulting exponential function plus one standard deviation (solid red line in Figs. 4 and 5).

In preparation of the following combined evaluation of the results from all tide gauges, the noise threshold functions from the different periodograms are averaged; separately for lunitidal intervals ($L_{\mathrm{i}}$) and heights ($L_{\mathrm{h}}$):

$$L_{\mathrm{i}}(\omega) = 0.0004816 \cdot \exp(-0.0101045\,\mathrm{tn}/^\circ \cdot \omega) \quad ,$$

$$L_{\mathrm{h}}(\omega) = 0.0024472 \cdot \exp(-0.0149899\,\mathrm{tn}/^\circ \cdot \omega) \quad .$$

These two functions represent mean lower intensity boundaries for the selection of significant peaks. The expressions $L_{\mathrm{i}}$ and $L_{\mathrm{h}}$ are unitless, due to the normalization of the Lomb-Scargle periodogram (Zechmeister and Kürster, 2009).

In addition to the intensity of a local maximum, the number of its occurrences in the different periodograms and its assignment to the partial tides determine the inclusion into the list of constituents for the HRoI. The local maxima must match the theoretically expectable partial tides that have well known angular velocities computable from the linear combinations of the rate of change of the four fundamental astronomical arguments $s$, $h$, $p$ and $N'$ (see Sect. 2). The angular velocities of 1268 partial tides have been precalculated using again the expressions for the fundamental astronomical arguments as published by the International Earth Rotation and Reference Systems Service (2010, Sect. 5.7). The ranges of the linear coefficients $m$ are chosen based on experience (compare Tab. 2):

$$m_s = 0,...,8$$

$$m_h = -8,...,3$$

$$m_p = -2,...,3, \quad \text{if } ((m_s = 0 \text{ and } m_h \geq 0)$$

$$\text{or } (m_s > 0 \text{ and } m_h \geq -m_s - 1))$$

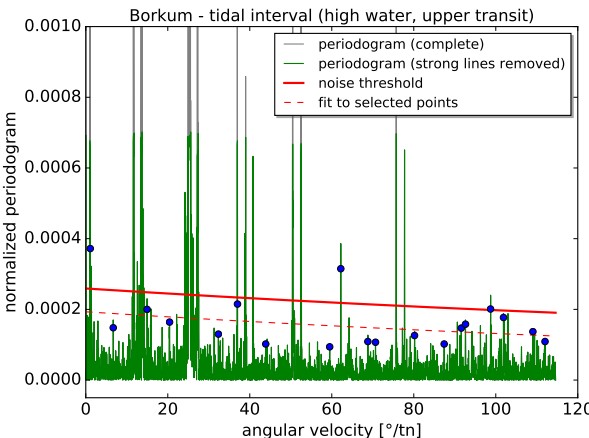

**Figure 4.** Determination of the noise threshold for the tidal interval (high water, upper transit) at tide gauge Borkum, Fischerbalje. The strongest lines are removed from the periodogram (grey vs. green lines; first step as described in Sect. 4.3) and an exponential function (dashed red curve) is fitted to selected points (blue; second step as described in Sect. 4.3). The noise threshold (thick red line) is shifted up by one standard deviation.

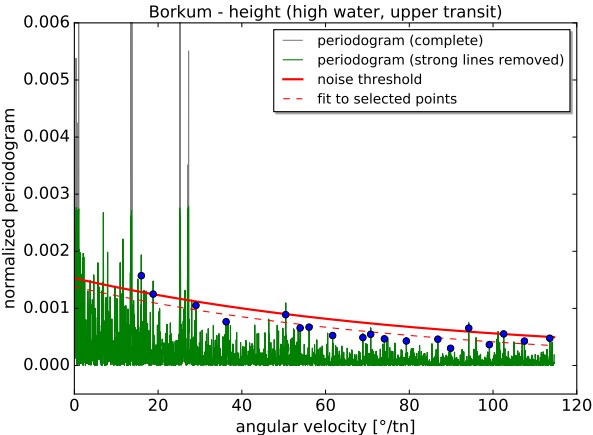

**Figure 5.** Same as Fig. 4 but for the heights at tide gauge Borkum, Fischerbalje.

$$m_{N'} = -1, 0, 1, \quad \text{if } ((m_s = 0 \text{ and } m_h = 0$$

$$\text{and } m_p = 0 \text{ and } m_{N'} \geq 0)$$

$$\text{or } (m_s \neq 0 \text{ or } m_h \neq 0$$

$$\text{or } m_p \neq 0))$$

A partial tide from the precalculated list is assigned uniquely to the closest peak in the periodogram if the difference in angular velocity is less than half the spectral resolution. The spectral resolution $r$ is defined as

$$r = 360°/T \quad , \tag{3}$$

with $T$ being the length of the time series in transit numbers. For example, the spectral resolution of a time series of 19 years is

$$r = \frac{360°}{19\,\text{yr} \cdot 365.25\,\text{d/yr} \cdot \tau} = 0.05°/\text{tn} \quad , \tag{4}$$

where $\tau = 1.03505013\,\text{d/tn}$ is the length of the mean lunar day.

For each identified partial tide, we calculate (i) the percentage of periodograms in which the partial tide has been detected, separately for lunitidal interval ($N_i$) and height ($N_h$); and (ii) the average intensity in the periodograms, separately for lunitidal interval ($I_i$) and height ($I_h$). In order to be considered relevant, a partial tide with angular velocity $\omega$ must meet the following criteria: $N_i \geq 33\%$ and $I_i(\omega) > L_i(\omega)$, or $N_h \geq 33\%$ and $I_h(\omega) > L_h(\omega)$. All partial tides that meet these selection criteria are listed Tab. 3.

**Table 3.** The most important partial tides that were identified in the periodograms, based on the combined evaluation of data from all tide gauges. See Sect. 4.3 for information on selection criteria and $I_i$, $I_h$, $N_i$ and $N_h$.

| Doodson | $\omega$ [°/tn] | $I_i$ [-] | $I_h$ [-] | $N_i$ [%] | $N_h$ [%] |
|---|---|---|---|---|---|
| ZZZZAZ | 0.0548098 | 0.0086 | 0.0102 | 78 | 45 |
| ZZZBZZ | 0.2306165 | 0.0019 | 0.0085 | 29 | 47 |
| ZZAXZZ | 0.7895780 | 0.0009 | 0.0088 | 16 | 34 |
| ZZAZZZ | 1.0201944 | 0.0070 | 0.0367 | 85 | 84 |
| ZZBZZZ | 2.0403886 | 0.0013 | 0.0068 | 26 | 56 |
| ZAXZZZ | 11.5978420 | 0.0009 | 0.0034 | 69 | 35 |
| ZAXAZZ | 11.7131503 | 0.0234 | 0.0024 | 100 | 10 |
| ZAYXZZ | 12.3874200 | 0.0006 | 0.0031 | 2 | 65 |
| ZAYZZZ | 12.6180365 | 0.0007 | 0.0042 | 34 | 21 |
| ZAYAAZ | 12.7881545 | 0.0005 | 0.0031 | 1 | 45 |
| ZAZYZZ | 13.5229227 | 0.0112 | 0.0119 | 99 | 90 |
| ZAZZZZ | 13.6382309 | 0.0105 | 0.0297 | 97 | 87 |
| ZAZAZZ | 13.7535391 | 0.0006 | 0.0016 | 63 | 2 |
| ZABBAZ | 15.9640460 | 0.0010 | 0.0032 | 1 | 70 |
| ZBWZZZ | 24.2158785 | 0.0029 | 0.0081 | 83 | 7 |
| ZBXZZZ | 25.2360729 | 0.4550 | 0.0706 | 100 | 92 |
| ZBYZZZ | 26.2562673 | 0.0034 | 0.0079 | 55 | 9 |
| ZBZYZZ | 27.1611535 | 0.0008 | 0.0021 | 43 | 29 |
| ZBZZZZ | 27.2764618 | 0.0382 | 0.0070 | 100 | 85 |
| ZCVAZZ | 36.9492232 | 0.0037 | 0.0013 | 99 | 25 |
| ZCXYZZ | 38.7589956 | 0.0009 | 0.0014 | 89 | 9 |
| ZCXAZZ | 38.9896120 | 0.0010 | 0.0004 | 96 | 0 |
| ZCZYZZ | 40.7993844 | 0.0008 | 0.0000 | 93 | 0 |
| ZDUZZZ | 49.4519514 | 0.0004 | 0.0000 | 38 | 0 |
| ZDVZZZ | 50.4721458 | 0.0196 | 0.0025 | 100 | 73 |
| ZDXZZZ | 52.5125347 | 0.0059 | 0.0015 | 99 | 15 |
| ZDZZZZ | 54.5529235 | 0.0003 | 0.0003 | 43 | 0 |
| ZETAZZ | 62.1852961 | 0.0006 | 0.0000 | 93 | 0 |

| Doodson | $\omega$ [°/tn] | $I_i$ [-] | $I_h$ [-] | $N_i$ [%] | $N_h$ [%] |
|---|---|---|---|---|---|
| continued from previous page | | | | | |
| ZEVYZZ | 63.9950685 | 0.0003 | 0.0001 | 61 | 0 |
| ZEVAZZ | 64.2256849 | 0.0004 | 0.0000 | 78 | 0 |
| ZFTZZZ | 75.7082187 | 0.0014 | 0.0003 | 98 | 2 |
| ZFVZZZ | 77.7486076 | 0.0010 | 0.0003 | 97 | 5 |
| ZHRZZZ | 100.9442917 | 0.0002 | 0.0000 | 36 | 0 |
| ZHTZZZ | 102.9846805 | 0.0002 | 0.0002 | 37 | 0 |

## 4.4 Adjustment of constituent list and ranking

In this section, we describe adjustments made to the list of partial tides based on manual inspections of certain periodograms and other considerations for an operational application. These adjustments lead to the set of partial tides in Tab. 4.

The periodograms calculated from longer time series offer a higher spectral resolution and contain more spectral information compared to the periodograms of shorter time series. This is demonstrated in the lower panels of Fig. 2 and 3 with periodograms based on time series from tide gauge Cuxhaven (115 years) and Emden (27 years). The higher information content from longer water level records needs to be appreciated and incorporated adequately. Therefore, the periodograms of Cuxhaven and Hamburg have been inspected manually to find partial tides that appear in the data of these two tide gauges and might not

be detectable in other periodograms. Six partial tides with the following Doodson numbers were identified and added to the list: ZAZZAZ (ZAZZZZ), ZBXZYZ (ZBXZZZ), ZBZXZZ, ZBZZAZ (ZBZZZZ), ZCXZZZ and ZDXZAZ (ZDXZZZ). The Doodson numbers in parenthesis are partial tides from Tab. 3 that differ only by $\Delta m_{N'} = \pm 1$. For these pairs, long time series are needed to clearly see two separate spectral lines in the periodograms.

    The noise in the periodograms increases towards lower angular velocities and the identification of partial tides below 1°/tn

becomes less clear. For this reason, and after inspecting several periodograms manually, the partial tide ZZAXZZ is considered to be a misidentification and has been removed from the list. Conversely, the partial tide ZZBXZZ has been added to the list, because of its importance for tide gauges located upstream in the Elbe river. Finally, we decided to cut the list after the eighth synodic month to keep the range of angular velocities consistent with previously used lists of partial tides (compare Tab. 2).

    The final set of long-period partial tides from our analysis is listed in Tab. 4. In the last column, each partial tide is assigned

a number $R$ indicating its overall importance (in decreasing order). The rank $R$ is based on the combined evaluation of data from all tide gauges and is calculated by the following procedure:

$$R_i = \text{rank}(\text{norm}(I_i(\omega) - L_i(\omega)) \cdot N_i)$$
$$R_h = \text{rank}(\text{norm}(I_h(\omega) - L_h(\omega)) \cdot N_h)$$
$$R = \text{rank}((3R_i + R_h)/4) \tag{5}$$

where the function norm() returns normalized values in the range [0,1] and the function rank() returns the position of a list element, if the list were sorted in increasing order. In Eq. 5, the results from lunitidal intervals are weighted stronger because the noise level is lower in the respective periodograms.

**Table 4.** The modified and adopted new list of long-period partial tides. The rank $R$ indicates the importance of a partial tide for tidal analysis, based on the combined evaluation of data from all tide gauges.

| Doodson | $\omega$ [°/tn] | description / name | $R$ |
|---|---|---|---|
| ZZZZAZ | 0.0548098 | lunar nodal precession | 6 |
| ZZZBZZ | 0.2306165 | half lunar apsidal precession | 13 |
| ZZAZZZ | 1.0201944 | tropical year / Sa | 7 |
| ZZBXZZ | 1.8097724 | | 31 |
| ZZBZZZ | 2.0403886 | half tropical year / Ssa | 17 |
| ZAXZZZ | 11.5978420 | | 14 |
| ZAXAZZ | 11.7131503 | MSm | 8 |
| ZAYXZZ | 12.3874200 | | 34 |
| ZAYZZZ | 12.6180365 | | 19 |
| ZAYAAZ | 12.7881545 | | 39 |
| ZAZYZZ | 13.5229227 | anomalistic month / Mm | 3 |
| ZAZZZZ | 13.6382309 | tropical month | 4 |
| ZAZZAZ | 13.6930407 | | 38 |
| ZAZAZZ | 13.7535391 | | 21 |
| ZABBAZ | 15.9640460 | | 36 |
| ZBWZZZ | 24.2158785 | | 11 |
| ZBXZYZ | 25.1812631 | | 35 |
| ZBXZZZ | 25.2360729 | half synodic month / MSf | 1 |
| ZBYZZZ | 26.2562673 | | 12 |
| ZBZXZZ | 27.0458453 | | 33 |
| ZBZYZZ | 27.1611535 | | 15 |
| ZBZZZZ | 27.2764618 | half tropical month / Mf | 2 |
| ZBZZAZ | 27.3312716 | | 27 |
| ZCVAZZ | 36.9492232 | S$\nu$2 | 10 |
| ZCXYZZ | 38.7589956 | SN | 16 |
| ZCXZZZ | 38.8743038 | | 24 |
| ZCXAZZ | 38.9896120 | MStm | 22 |
| ZCZYZZ | 40.7993844 | Mfm | 23 |
| ZDUZZZ | 49.4519514 | | 29 |
| ZDVZZZ | 50.4721458 | fourth synodic month / 2SM | 5 |
| | | continued on next page | |

| Doodson | $\omega$ [°/tn] | description / name | $R$ |
|---------|-----------------|-------------------|-----|
| | continued from previous page | | |
| ZDXZZZ | 52.5125347 | MSqm | 9 |
| ZDXZAZ | 52.5673444 | | 37 |
| ZDZZZZ | 54.5529235 | | 30 |
| ZETAZZ | 62.1852961 | | 25 |
| ZEVYZZ | 63.9950685 | 2SMN | 28 |
| ZEVAZZ | 64.2256849 | | 26 |
| ZFTZZZ | 75.7082187 | sixth synodic month | 20 |
| ZFVZZZ | 77.7486076 | | 18 |
| ZHRZZZ | 100.9442917 | eighth synodic month | 32 |

The rank $R$ can be used to select a sublist of partial tides when performing a tidal analysis of water levels with less than 18.6 years of data. This is important, because not all partial tides can be resolved against each other for shorter time series. The minimum difference in angular velocity is given by the resolution criterion (Eq. 3). Figure 6 illustrates the resolvable partial tides as a function of the length of the data record. Note that the high-ranked partial tide representing the tropical month which occurs at $R = 4$ in the list cannot be included for time series shorter than about nine years. For a tidal analysis of time series shorter than nine years, it is therefore often better to perform a reference analysis: 19 years of data are used from a different tide gauge with a similar tidal behaviour (e.g. similar course of the semi-monthly inequality) and the results are translated to the original gauge by applying the respective differences of the mean lunitidal intervals and mean heights. This way, nodal corrections can be avoided which come with their own assumptions and approximations (e.g. Godin, 1986; Amin, 1987).

## 5 Comparison of predictions with observations: two different lists of constituents for the HRoI

For verification of the new constituent list, tidal predictions based on an existing list of partial tides and based on the new set are compared with observations. The predictions are made for the year 2016 and are compared with tide gauge observations from the same year.

### 5.1 Tidal analysis and prediction

We calculate tidal predictions (times and heights of high and low waters) with the HRoI using (i) the 43 partial tides from "set 2" in Tab. 2 and (ii) the 39 partial tides derived from our analysis. The data and software are otherwise identical. The predictions are based on amplitudes $a_m$ (see Eq. 1) that are determined from a tidal analysis of water level records from 1995 to 2013 (19 years). The analysis is applied in two iterations with a 3-sigma-clipping in between. Data are only used from tide gauges that have delivered enough observations in this time period to include all partial tides in the analysis. Additionally, the tide gauges must have deliver observations for the year 2016. The 98 tide gauges that fulfil these criteria are marked in the column "used for verif." in Tab. A1. All tide gauge data is prepared as described in Sect. 4.1, including the removal of extreme events from the observations that are used for comparison.

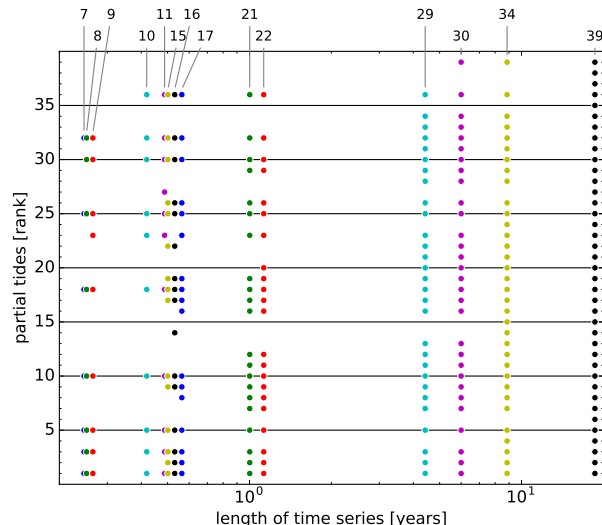

**Figure 6.** The partial tides (identified by their rank $R$ from Tab. 4) that can be resolved as a function of the (minimum) length of the time series. If two partial tides cannot be resolved against each other, the one with the lower rank is dropped. Note the logarithmic time axis from 0.2 to 20 years. The numbers at the top are the counts of partial tides.

## 5.2 Evaluation of residuals

In this section, we present results from the analysis of the residuals in the following order: the distributions of residuals for the tide gauge Cuxhaven, the means and standard deviations for some major ports, the changes in the standard deviations for all tide gauges, and the changes in the remaining frequencies. The residuals are the differences between the observed and the predicted vertices (times and heights of high and low waters) with the same assigned transit number and event index $k$.

Figure 7 shows histograms of the residuals for the tide gauge Cuxhaven. The panels on the left and on the right display histograms for the times and heights, respectively. Each panel contains one histogram for the tidal prediction with 43 partial tides (red) and one histogram for the tidal prediction with 39 partial tides (yellow). Using the new set of partial tides, the standard deviation of the residuals decreases from 9.6 min to 9.0 min for the times and from 0.28 m to 0.27 m for the heights.

In the same way as for Cuxhaven, residuals are calculated for the data of all 98 tide gauges included in the verification. The mean values $\mu$ and standard deviations $\sigma$ of the residuals for eleven major ports are summarized in Tab. 5 for the times and in Tab. 6 for the heights. Based on the results from all tide gauges, the average standard deviation of the residuals is 13.2 min or 0.28 m, respectively, using the set of 39 tidal constituents. In most cases, the new set of partial tides gives small improvements in $\mu$ and $\sigma$.

**Table 5.** Residuals of predicted and observed times of high and low water: mean $\mu$ and standard deviation $\sigma$ in minutes.

| gauge number | gauge name | 43 partial tides | | 39 partial tides | |
|---|---|---|---|---|---|
| | | $\mu$ | $\sigma$ | $\mu$ | $\sigma$ |
| 101P | Borkum | −2.7 | 11.2 | −2.1 | 11.0 |
| 103P | Bremerhaven | −6.4 | 10.4 | −4.6 | 10.1 |
| 111P | Norderney | 1.5 | 10.9 | 0.7 | 10.6 |
| 502P | Bremen | −7.3 | 10.8 | −5.5 | 10.5 |
| 505P | Büsum | 3.4 | 17.5 | 4.6 | 17.4 |
| 506P | Cuxhaven | −0.1 | 9.6 | 1.0 | 9.0 |
| 507P | Emden | −9.2 | 13.8 | −8.2 | 13.3 |
| 508P | Hamburg | −10.1 | 10.4 | −7.7 | 10.3 |
| 509A | Helgoland | −2.4 | 7.8 | −2.6 | 7.7 |
| 510P | Husum | −5.1 | 12.0 | −4.3 | 11.7 |
| 512P | Wilhelmshaven | −3.2 | 10.0 | −2.7 | 9.6 |

**Table 6.** Residuals of predicted and observed heights of high and low water: mean $\mu$ and standard deviation $\sigma$ in meters.

| gauge number | gauge name | 43 partial tides | | 39 partial tides | |
|---|---|---|---|---|---|
| | | $\mu$ | $\sigma$ | $\mu$ | $\sigma$ |
| 101P | Borkum | 0.05 | 0.24 | 0.03 | 0.24 |
| 103P | Bremerhaven | 0.06 | 0.28 | 0.04 | 0.28 |
| 111P | Norderney | 0.03 | 0.25 | −0.01 | 0.24 |
| 502P | Bremen | 0.01 | 0.27 | −0.01 | 0.26 |
| 505P | Büsum | 0.04 | 0.29 | 0.01 | 0.28 |
| 506P | Cuxhaven | 0.05 | 0.28 | 0.04 | 0.27 |
| 507P | Emden | −0.01 | 0.28 | −0.02 | 0.27 |
| 508P | Hamburg | −0.08 | 0.33 | −0.12 | 0.31 |
| 509A | Helgoland | 0.03 | 0.25 | 0.02 | 0.24 |
| 510P | Husum | 0.02 | 0.29 | 0.00 | 0.28 |
| 512P | Wilhelmshaven | 0.05 | 0.27 | 0.03 | 0.27 |

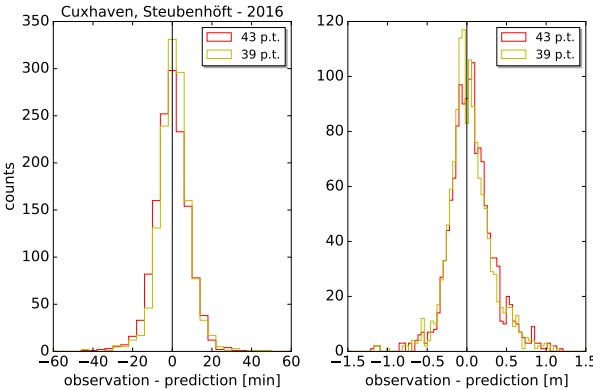

**Figure 7.** Histograms of the residuals for tide gauge Cuxhaven and year 2016. The different colours indicate predictions based on the different sets of partial tides (red: predictions with 43 partial tides; yellow: predictions with 39 partial tides). Left: time differences with a bin width of 4 min. Right: height differences with a bin width of 0.04 m.

The percentage changes $\Delta\sigma$ of the standard deviations are presented in the histograms in Fig. 8 (times) and Fig. 9 (heights) for all 98 tide gauges. The percentage change is calculated in the following way:

$$\Delta\sigma[\%] = 100\% \cdot \frac{\sigma_{39 \text{ p.t.}} - \sigma_{43 \text{ p.t.}}}{\sigma_{43 \text{ p.t.}}} \quad , \tag{6}$$

where $\sigma_{39 \text{ p.t.}}$ and $\sigma_{43 \text{ p.t.}}$ are the standard deviations of the residuals using the predictions with 39 partial tides and 43 partial tides, respectively. The average reductions of the standard deviations are 2.41% (times) and 2.30% (heights). The two tide gauges with the largest improvements in this study, both with regard to times and heights, are Holmer Siel (BSH gauge number 649B) at the North Frisian coast and Bremervörde (687P) in the river Oste. Further tide gauges with improved standard deviations are located all around the area of investigation. Regarding the times (Fig. 8), the four tide gauges with increased standard deviations are Westerland (620P) and Hörnum (624P) located at the North Frisian island of Sylt, and Dove-Elbe (727P) and Bunthaus (729P) located upstream the river Elbe. Regarding the heights (Fig. 9), the five tide gauges with the largest positive changes are also located upstream the river Elbe, namely Geesthacht (732D), Altengamme (732A), Zollenspieker (731P), Illmenau (730A) and Fahrenholz (730C). The water levels in this part of the Elbe are partly influenced by river discharge, which can lead to deviations from the tidal predictions.

The change of constituents has an influence on the remaining periodicities in the residuals. Periodograms are calculated for the two sets of residuals (times and heights) for each tide gauge. The 98 periodograms of each type are averaged. The resulting mean periodograms are shown in Fig. 10 and 11. In both figures, the strongest peaks are located at very low angular velocities ($\lesssim 1°$/tn). As mentioned before, the unambiguous identification of partial tides at these periods is difficult and consequently no major improvements are achieved in reducing the (average) residual periodicities in this range. Four further strong peaks are visible in both figures at about 15, 25, 52 and 64°/tn for the prediction with 43 partial tides. These peaks are clearly reduced with the new predictions (39 partial tides).

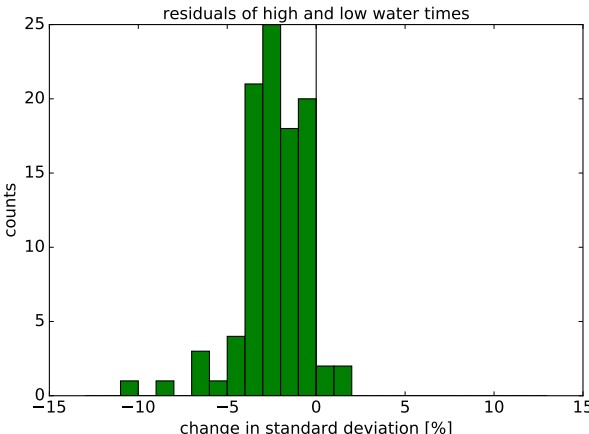

**Figure 8.** Histogram of the change in the standard deviation of the residuals of high and low water times for all 98 tide gauges.

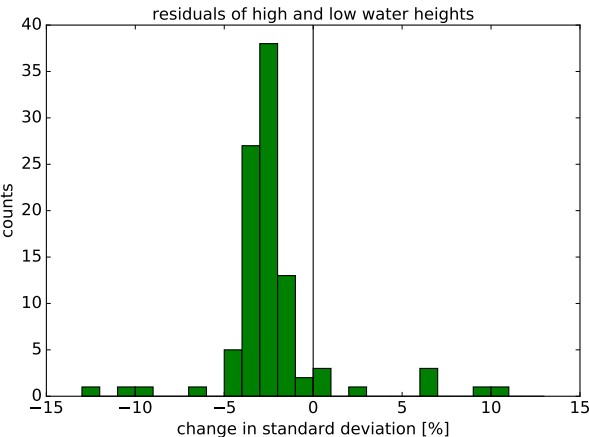

**Figure 9.** Histogram of the change in the standard deviation of the residuals of high and low water heights for all 98 tide gauges.

## 6   Comparison of predictions with observations: the HRoI and the harmonic method

The harmonic method is the most widely used technique for tidal predictions. The following comparison of predictions calculated with the HRoI and with the harmonic method shall demonstrate the respective capabilities. The comparison is done for the two tide gauges at Cuxhaven, Steubenhöft, and Hamburg, St. Pauli. The first site is located at the mouth of the river Elbe into the North Sea, while the second is about 100 km upstream in the river Elbe. The predictions are compared with tide gauge observations from the year 2016.

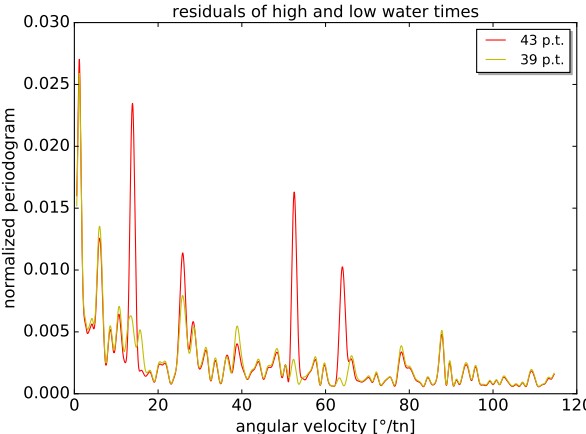

**Figure 10.** Mean periodogram of residual high and low water times for all tide gauges used in the verification. The different colours indicate predictions based on the different sets of partial tides (red: predictions with 43 partial tides; yellow: predictions with 39 partial tides)

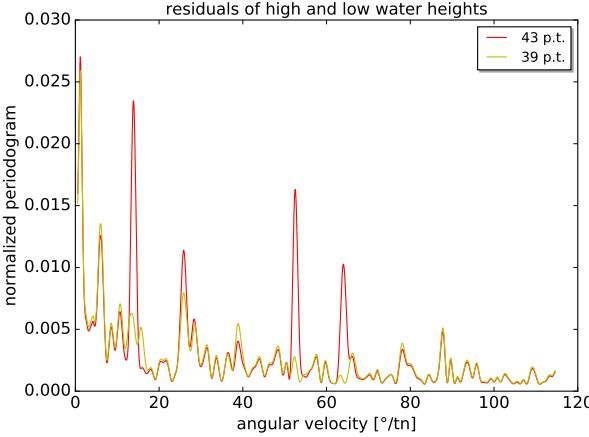

**Figure 11.** Same as Fig. 10, but for the high and low water heights.

## 6.1 Tidal analysis and prediction

The predictions with the HRoI (39 partial tides) are the same as in Sect. 5. The harmonic analysis is based on continuous observations from the years 1996-2014 at 10-minute intervals. The harmonic constituents (amplitudes $H$ and phases $g$) and the constant vertical offset $Z_0$ are determined from a least-squares fit with the following model function:

$$\hat{y}_{\mathrm{harm}} = Z_0 + \sum_{l=1}^{L} [H_l \cdot \cos(V_l(t_0) + \omega_l t - g_l)] \quad . \tag{7}$$

We use the 68 partial tides (with angular velocities $\omega_l$) from Foreman (1977). These are also the default constituents in the Matlab packages t_tide and UTide (Pawlowicz et al., 2002; Codiga, 2011) which have become widely accepted standard implementations of the harmonic method. Since the data records exceed 18.6 years, we add the partial tide with the angular velocity of the lunar node and omit nodal corrections. This gives a total of $L = 69$ constituents. The time $t$ is referenced to the midpoint $t_0$ of the time series and the astronomical argument $V_l(t_0)$ is calculated for each partial tide using the expressions for the fundamental astronomical arguments as published by the International Earth Rotation and Reference Systems Service (2010, Sect. 5.7). The analysis is applied in two iterations with a 3-sigma-clipping in between to remove outliers in the observations.

We show in Figs. 12 and 13 the predictions and observations for Cuxhaven and Hamburg, respectively. Only ten days in June 2016 are shown from the complete time series for better visibility of the individual high and low waters. The two curves in each figure are the observed water levels (dark blue) and the harmonic prediction (light green). The high and low waters are marked separately for observations (red circles), vertices determined from the harmonic prediction (green squares) and predictions made with the HRoI (yellow triangles).

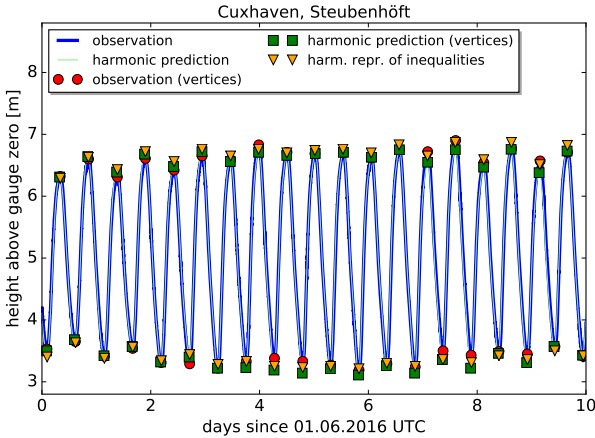

**Figure 12.** Observations and two predictions for the tide gauge Cuxhaven, Steubenhöft. The first 10 days of June 2016 are shown.

## 6.2 Evaluation of residuals

As in Sect. 5, the residuals are the differences between the observed and the predicted vertices (times and heights of high and low waters) with the same assigned transit number and event index $k$. We calculate the means and the standard deviations of the residuals regarding times and heights. The results are shown in Tab. 7. The differences for the heights are within a few centimetres. For the times, the standard deviations are approximately $4 - 5$ minutes larger in the case of the harmonic method. The residuals for the times are also shown in Fig. 14, where the curves for the harmonic method (blue) suggest that long-period periodicities could be present in the residuals which are not covered by the predictions. Based on the calculated parameters, the deviations of the harmonic prediction from the observations (and also from the prediction made with the HRoI) are larger

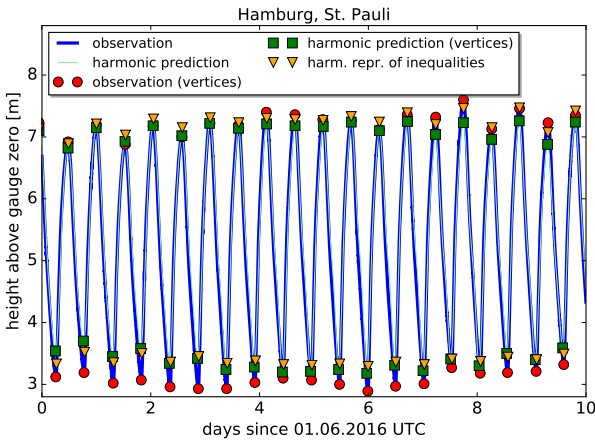

**Figure 13.** Observations and two predictions for the tide gauge Hamburg, St. Pauli. The first 10 days of June 2016 are shown.

**Table 7.** Residuals of predicted and observed times and heights of high and low water: mean $\mu$ and standard deviation $\sigma$.

| | times [min] | | | |
|---|---|---|---|---|
| gauge name | HRoI (39 p.t.) | | harmonic pred. | |
| | $\mu$ | $\sigma$ | $\mu$ | $\sigma$ |
| Cuxhaven | 1.0 | 9.0 | 12.0 | 12.9 |
| Hamburg | −7.7 | 10.3 | 11.8 | 15.1 |

| | heights [m] | | | |
|---|---|---|---|---|
| gauge name | HRoI (39 p.t.) | | harmonic pred. | |
| | $\mu$ | $\sigma$ | $\mu$ | $\sigma$ |
| Cuxhaven | 0.03 | 0.27 | 0.05 | 0.29 |
| Hamburg | −0.11 | 0.31 | −0.10 | 0.37 |

for Hamburg as compared to Cuxhaven. This supports the assumptions that the application of the HRoI is especially useful for tide gauge locations in rivers.

## 7 Conclusions

Time series of high and low water records from 111 German tide gauges were analysed to determine important long-period
5   partial tides. Generalized Lomb-Scargle periodograms were calculated from lunitidal intervals and heights for all tide gauges, and spectral peaks were identified in these periodograms above noise thresholds. The separate analyses of lunitidal intervals

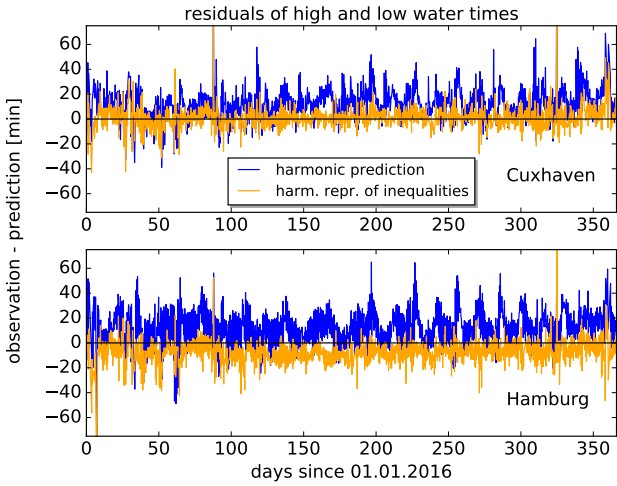

**Figure 14.** The residuals of high and low water times for the tide gauges Cuxhaven (upper panel) and Hamburg (lower panel).

and heights were combined to realize one comprehensive list of partial tides. An application is the usage of these constituents in tidal analyses and predictions with the HRoI.

The new set of 39 partial tides largely confirms the previously used set with 43 partial tides. It can be seen from Tab. 2, that nine new constituents were added and 13 constituents were removed. Many of the removed angular velocities are close to strong partial tides, such as the anomalistic month (Mm), the half synodic month (MSf) and the half tropical month (Mf). The removed constituents might have been artefacts from spectral leakage, which are most prominent in the proximity of strong spectral lines and which were misidentified as true signals in previous studies. The unambiguous identification of partial tides is very difficult at angular velocities below approximately 1°/tn because the noise levels in the periodograms increase towards lower angular velocities and the results from different tide gauges are less consistent.

The verification based on observations from 98 tide gauges in the year 2016 suggests that the usage of the new constituents list can lead to slightly better predictions. In particular, the average standard deviations of the residuals are lower and four frequencies were reduced.

This study presents for the first time a thorough investigations of the long-period constituents used with the HRoI. The new list of constituents will be used in tidal analyses and predictions with the HRoI for German tide gauges starting with the BSH tide tables for the year 2020.

In future work, extensive comparison of the HRoI with the common harmonic method might provide more insights into the capabilities of both tidal analysis techniques. The German Bight would be an ideal area of investigation with its large network of tide gauges located both at the open North Sea and far within tidally influenced rivers.

## Appendix A: Table of tide gauges

**Table A1.** 137 German tide gauges which deliver water level observations on a regular basis and for which tidal predictions were published in BSH tide tables (*Gezeitentafeln*) or tide calendar (*Gezeitenkalender*) for the year 2018. The data from the tide gauges are observed times and heights of high and low water. The tide gauges are operated by different federal and state agencies which provide tide gauge records to BSH. Abbreviations in the third column correspond to the following agencies: E: Emden Waterways and Shipping Authority (Wasserstraßen- und Schifffahrtsamt Emden, WSA Emden), BH: WSA Bremerhaven, B: WSA Bremen, C: WSA Cuxhaven, T: WSA Tönning, HPA: Hamburg Port Authority, W: WSA Wilhelmshaven, HU: Landesbetrieb für Küstenschutz, Nationalpark und Meeresschutz Schleswig-Holstein (LKN-SH Husum), H: WSA Hamburg, L: WSA Lauenburg, N: Niedersächsischer Landesbetrieb für Wasserwirtschaft, Küsten- und Naturschutz (NLWKN), M: WSA Meppen.

| BSH gauge number | gauge name | auth. | data period [start/end date] | data period [years] | completeness of data [%] | used for analysis | used for verif. |
|---|---|---|---|---|---|---|---|
| 101P | Borkum, Fischerbalje | E | 02.01.1963-31.12.2015 | 53.0 | 62 | x | x |
| 103P | Bremerhaven, Alter Leuchtturm | BH | 01.11.1965-31.12.2015 | 50.2 | 62 | x | x |
| 111P | Norderney, Riffgat | E | 01.01.1964-31.12.2015 | 52.0 | 67 | x | x |
| 502P | Bremen, Oslebshausen | B | 01.01.1950-31.12.2015 | 66.0 | 99 | x | x |
| 504A | Brunsbüttel, Mole 1 | C | 01.08.2010-31.12.2015 | 5.4 | 95 | | |
| 505P | Büsum | T | 01.01.1963-31.12.2015 | 53.0 | 63 | x | x |
| 506P | Cuxhaven, Steubenhöft | C | 01.01.1901-31.12.2015 | 115.0 | 99 | x | x |
| 507P | Emden, Große Seeschleuse | E | 01.01.1989-31.12.2015 | 27.0 | 99 | x | x |
| 508P | Hamburg, St. Pauli | HPA | 01.01.1901-31.12.2015 | 115.0 | 100 | x | x |
| 509A | Helgoland, Binnenhafen | T | 01.01.1989-31.12.2015 | 27.0 | 99 | x | x |
| 510P | Husum | T | 01.01.1989-31.12.2015 | 27.0 | 98 | x | x |
| 512P | Wilhelmshaven, Alter Vorhafen | W | 01.01.1973-31.12.2015 | 43.0 | 98 | x | x |
| 613C | Hojer, Schleuse | HU | 01.01.1999-31.12.2015 | 17.0 | 100 | | |
| 617P | List, Hafen | T | 01.01.1986-31.12.2015 | 30.0 | 98 | x | x |
| 618P | Munkmarsch | HU | 16.01.1989-31.12.2015 | 27.0 | 49 | | |
| 620P | Westerland | HU | 01.01.1986-31.12.2015 | 30.0 | 94 | x | x |
| 622P | Amrum Odde | HU | 17.04.1996-07.12.2015 | 19.6 | 40 | | |
| 623A | Rantumdamm | HU | 08.01.1996-31.12.2015 | 20.0 | 89 | x | |
| 624P | Hörnum, Hafen | T | 01.01.1989-31.12.2015 | 27.0 | 99 | x | x |
| 628A | Osterley | HU | 09.04.1997-11.11.2015 | 18.6 | 40 | | |
| 629B | Föhrer Ley Nord | HU | 27.04.1994-11.11.2015 | 21.5 | 46 | | |
| 631P | Amrum, Hafen (Wittdün) | T | 01.01.1989-31.12.2015 | 27.0 | 96 | x | |
| 632P | Föhr, Wyk | HU | 01.01.1989-31.12.2015 | 27.0 | 100 | x | x |

| BSH gauge number | gauge name | auth. | data period [start/end date] | data period [years] | completeness of data [%] | used for analysis | used for verif. |
|---|---|---|---|---|---|---|---|
| 635P | Dagebüll | T | 01.01.1989-31.12.2015 | 27.0 | 99 | x | x |
| 636F | Hooge, Anleger | HU | 01.01.1989-31.12.2015 | 27.0 | 93 | x | x |
| 637A | Strand, Hamburger Hallig | HU | 02.05.1989-31.12.2015 | 26.7 | 79 | x | |
| 637P | Gröde, Anleger | HU | 01.01.1989-31.12.2015 | 27.0 | 48 | | |
| 638P | Schlüttsiel | HU | 01.01.1989-31.12.2015 | 27.0 | 96 | x | x |
| 642C | Rummelloch, West | HU | 14.06.1994-08.12.2015 | 21.5 | 47 | | |
| 645P | Süderoogsand | HU | 23.03.1993-20.11.2015 | 22.7 | 62 | x | |
| 647A | Pellworm, Anleger | T | 01.03.1996-31.12.2015 | 19.8 | 88 | x | |
| 649B | Holmer Siel | HU | 01.01.1994-31.12.2015 | 22.0 | 89 | x | x |
| 649P | Nordstrand, Strucklahnungshörn | HU | 01.01.1989-31.12.2015 | 27.0 | 96 | x | x |
| 653P | Südfall, Fahrwasserkante | HU | 25.03.1993-02.12.2015 | 22.7 | 67 | x | |
| 655D | Tümlauer Hafen | HU | 24.09.2001-31.12.2013 | 12.3 | 93 | | |
| 658B | Linnenplate | HU | 11.04.2001-05.12.2013 | 12.7 | 62 | | |
| 664P | Eidersperrwerk, AP | T | 01.01.1989-31.12.2014 | 26.0 | 97 | x | |
| 666P | Blauort | HU | 12.01.1989-31.12.2015 | 27.0 | 82 | x | x |
| 667B | Meldorf - Sperrwerk, AP | HU | 04.01.1994-31.12.2015 | 22.0 | 71 | x | |
| 669P | Deichsiel | HU | 01.01.1989-31.12.2013 | 25.0 | 96 | x | |
| 673P | Trischen, West | HU | 18.03.1989-25.11.2015 | 26.7 | 61 | x | |
| 675C | Mittelplate | HU | 01.01.1992-25.11.2015 | 23.9 | 55 | | |
| 675P | Friedrichskoog, Hafen | HU | 01.01.1989-31.12.2015 | 27.0 | 100 | x | x |
| 676P | Zehnerloch | C | 01.01.1989-14.11.2015 | 26.9 | 98 | x | x |
| 677C | Scharhörnriff, Bake A | C | 01.01.2001-31.12.2015 | 15.0 | 96 | | |
| 677P | Scharhörn, Bake C | C | 01.01.1989-31.12.2015 | 27.0 | 98 | x | x |
| 678W | Neuwerk | HPA | 01.01.1994-31.12.2015 | 22.0 | 46 | | |
| 681A | Neufeld, Hafen | HU | 01.01.1994-31.12.2015 | 22.0 | 78 | x | x |
| 681P | Otterndorf | C | 01.01.1989-31.12.2015 | 27.0 | 79 | x | |
| 682P | Osteriff | C | 01.01.1989-31.12.2015 | 27.0 | 89 | x | |
| 683P | Belum, Oste | C | 01.01.1989-31.12.2015 | 27.0 | 96 | x | x |
| 685P | Hechthausen, Oste | C | 01.01.1989-31.12.2015 | 27.0 | 95 | x | x |
| 687P | Bremervörde, Oste | C | 01.01.1989-31.12.2015 | 27.0 | 94 | x | x |
| 688P | Brokdorf | H | 01.01.1989-31.12.2015 | 27.0 | 98 | x | x |
| 690P | Stör - Sperrwerk, AP | HU | 01.01.2000-31.12.2015 | 16.0 | 87 | | |
| 691R | Kasenort, Stör | HU | 01.01.1989-31.12.2015 | 27.0 | 99 | x | x |

| BSH gauge number | gauge name | auth. | data period [start/end date] | data period [years] | completeness of data [%] | used for analysis | used for verif. |
|---|---|---|---|---|---|---|---|
| 692P | Itzehoe, Stör | H | 01.01.1989-31.12.2015 | 27.0 | 97 | x | |
| 693P | Breitenberg, Stör | H | 01.01.2000-31.12.2015 | 16.0 | 95 | | |
| 695P | Glückstadt | H | 01.01.1989-31.12.2015 | 27.0 | 95 | x | x |
| 697P | Krautsand | H | 01.01.1989-31.12.2015 | 27.0 | 89 | x | x |
| 698P | Kollmar (Kamperreihe) | H | 01.01.1989-31.12.2015 | 27.0 | 98 | x | x |
| 700R | Krückau - Sperrwerk, BP | H | 01.01.2000-31.12.2015 | 16.0 | 89 | | |
| 703P | Grauerort | H | 01.01.1989-31.12.2015 | 27.0 | 98 | x | x |
| 704R | Pinnau - Sperrwerk, BP | H | 01.01.2000-31.12.2015 | 16.0 | 93 | | |
| 706P | Uetersen, Pinnau | H | 01.01.1989-31.12.2015 | 27.0 | 92 | x | x |
| 709P | Stadersand, Schwinge | H | 01.01.1989-31.12.2015 | 27.0 | 98 | x | x |
| 711P | Hetlingen | H | 01.01.1989-31.12.2015 | 27.0 | 94 | x | x |
| 712P | Lühort, Lühe | H | 01.01.1989-31.12.2015 | 27.0 | 97 | x | x |
| 714P | Schulau | H | 01.01.1989-31.12.2015 | 27.0 | 97 | x | x |
| 715P | Blankenese, Unterfeuer | HPA | 01.01.1989-31.12.2015 | 27.0 | 98 | x | x |
| 717P | Cranz, Este - Sperrwerk, AP | H | 01.01.1989-31.12.2015 | 27.0 | 83 | x | x |
| 718P | Buxtehude, Este | H | 01.01.1989-31.12.2015 | 27.0 | 86 | x | x |
| 720P | Seemannshöft | HPA | 01.01.1989-31.12.2015 | 27.0 | 100 | x | x |
| 724P | Harburg, Schleuse | HPA | 01.01.1989-31.12.2015 | 27.0 | 100 | x | x |
| 727P | Dove - Elbe, Einfahrt | HPA | 01.01.1989-31.12.2015 | 27.0 | 99 | x | x |
| 729P | Bunthaus | HPA | 01.01.1989-31.12.2015 | 27.0 | 100 | x | x |
| 730A | Ilmenau - Sperrwerk, AP | L | 01.01.1989-31.12.2015 | 27.0 | 98 | x | x |
| 730C | Fahrenholz, Ilmenau | L | 01.01.1989-31.12.2015 | 27.0 | 93 | x | x |
| 730P | Over | L | 01.01.1989-31.12.2015 | 27.0 | 98 | x | x |
| 731P | Zollenspieker | L | 01.01.1989-31.12.2015 | 27.0 | 98 | x | x |
| 732A | Altengamme | L | 01.01.1989-31.12.2015 | 27.0 | 96 | x | x |
| 732D | Geesthacht, Wehr UP | L | 01.01.1989-31.12.2015 | 27.0 | 98 | x | x |
| 734P | Alte Weser, Leuchtturm | BH | 01.01.1989-31.12.2015 | 27.0 | 99 | x | x |
| 735A | Spieka Neufeld | N | 01.01.1989-31.12.2015 | 27.0 | 50 | | |
| 735B | Wremertief | N | 01.01.1994-31.12.2015 | 22.0 | 38 | | |
| 737P | Dwarsgat, Unterfeuer | BH | 01.01.1989-31.12.2015 | 27.0 | 99 | x | x |
| 737S | Robbensüdsteert | BH | 01.01.1989-31.12.2015 | 27.0 | 96 | x | x |
| 738P | Fedderwardersiel | N | 01.01.1989-31.12.2015 | 27.0 | 49 | | |
| 741A | Nordenham, Unterfeuer | BH | 01.01.1989-31.12.2015 | 27.0 | 99 | x | x |

| BSH gauge number | gauge name | auth. | data period [start/end date] | data period [years] | completeness of data [%] | used for analysis | used for verif. |
|---|---|---|---|---|---|---|---|
| 741B | Rechtenfleth | BH | 01.01.1993-31.12.2015 | 23.0 | 99 | x | x |
| 743P | Brake | B | 01.01.1989-31.12.2015 | 27.0 | 97 | x | x |
| 744A | Elsfleth Ohrt | B | 01.01.1989-31.12.2015 | 27.0 | 96 | x | x |
| 744P | Elsfleth | B | 01.01.1975-31.12.2015 | 41.0 | 80 | x | x |
| 745P | Huntebrück, Hunte | B | 01.01.1989-31.12.2015 | 27.0 | 98 | x | x |
| 746P | Hollersiel, Hunte | B | 01.01.1989-31.12.2015 | 27.0 | 98 | x | x |
| 747P | Reithörne, Hunte | B | 01.01.1989-31.12.2015 | 27.0 | 98 | x | x |
| 748P | Oldenburg - Drielake, Hunte | B | 01.01.1989-31.12.2015 | 27.0 | 97 | x | x |
| 749P | Farge | B | 01.01.1989-31.12.2015 | 27.0 | 99 | x | x |
| 750A | Wasserhorst, Lesum | B | 01.01.1989-31.12.2015 | 27.0 | 90 | x | x |
| 750B | Ritterhude, Hamme | B | 01.01.1989-31.12.2015 | 27.0 | 91 | x | x |
| 750C | Niederblockland, Wümme | B | 01.01.1989-31.12.2015 | 27.0 | 91 | x | x |
| 750D | Borgfeld, Wümme | B | 01.01.1989-31.12.2015 | 27.0 | 90 | x | x |
| 750P | Vegesack | B | 01.01.1975-31.12.2015 | 41.0 | 80 | x | x |
| 751P | Bremen, Wilhelm-Kaisen-Brück | B | 01.01.1989-31.12.2015 | 27.0 | 99 | x | x |
| 752P | Bremen, Weserwehr | B | 01.01.1989-31.12.2015 | 27.0 | 98 | x | x |
| 754P | Wangerooge, Langes Riff, (Nord) | W | 01.01.1976-31.12.2015 | 40.0 | 71 | x | x |
| 756P | Wangerooge, Ost | W | 01.05.1976-31.12.2015 | 39.7 | 58 | | x |
| 760P | Mellumplate, Leuchtturm | W | 01.01.1989-31.12.2015 | 27.0 | 99 | x | x |
| 761P | Schillig | W | 01.01.1989-31.12.2015 | 27.0 | 92 | x | x |
| 764B | Hooksielplate | W | 01.01.1989-31.12.2015 | 27.0 | 93 | x | x |
| 766P | Voslapp | W | 01.01.1989-31.12.2015 | 27.0 | 95 | x | x |
| 769P | Wilhelmshaven, Ölpier | W | 01.01.1989-31.12.2015 | 27.0 | 97 | x | x |
| 770P | Wilhelmshaven, Neuer Vorhafen | W | 01.01.1989-31.12.2015 | 27.0 | 97 | x | x |
| 773P | Arngast, Leuchtturm | W | 15.05.2001-31.12.2015 | 14.6 | 88 | | |
| 776P | Vareler Schleuse | N | 01.01.1989-31.12.2015 | 27.0 | 49 | | |
| 777P | Wangerooge, West | W | 01.01.1976-31.12.2015 | 40.0 | 73 | x | x |
| 778P | Harlesiel | N | 01.01.1989-31.12.2015 | 27.0 | 57 | | |
| 779P | Spiekeroog | E | 01.01.1989-31.12.2015 | 27.0 | 98 | x | x |
| 781P | Langeoog | E | 01.01.1989-31.12.2015 | 27.0 | 98 | x | x |
| 782P | Bensersiel | N | 01.01.1989-31.12.2015 | 27.0 | 100 | x | x |
| 796C | Leybucht, Leyhörn | N | 01.01.1992-31.12.2015 | 24.0 | 100 | x | x |
| 798P | Borkum, Südstrand | E | 02.01.1989-31.12.2015 | 27.0 | 95 | x | x |

| BSH gauge number | gauge name | auth. | data period [start/end date] | data period [years] | completeness of data [%] | used for analysis | used for verif. |
|---|---|---|---|---|---|---|---|
| 799G | Dukegat | E | 01.01.1989-31.12.2015 | 27.0 | 95 | x | x |
| 799P | Emshörn | E | 01.01.1989-31.12.2015 | 27.0 | 99 | x | x |
| 802P | Knock | E | 01.01.1989-31.12.2015 | 27.0 | 99 | x | x |
| 803P | Pogum, Ems | E | 01.01.1989-31.12.2015 | 27.0 | 98 | x | x |
| 805P | Terborg, Meßstelle, Ems | E | 01.01.1989-31.12.2015 | 27.0 | 98 | x | x |
| 806P | Leerort, Ems | E | 01.01.1989-31.12.2015 | 27.0 | 97 | x | x |
| 808A | Leda - Sperrwerk, Unterpegel | E | 01.01.1989-31.12.2015 | 27.0 | 98 | x | x |
| 810A | Nortmoor, Altarm Jümme | N | 01.01.2000-31.12.2015 | 16.0 | 85 | | |
| 810B | Detern, Jümme | N | 01.01.2000-31.12.2015 | 16.0 | 86 | | |
| 810P | Westringaburg, Leda | N | 01.01.1989-31.12.2015 | 27.0 | 84 | x | x |
| 812P | Dreyschloot, Leda | E | 01.01.1989-31.12.2015 | 27.0 | 89 | x | x |
| 813P | Weener, Ems | E | 01.01.1989-31.12.2015 | 27.0 | 98 | x | x |
| 814B | Rhede, Ems | M | 01.01.1989-31.12.2015 | 27.0 | 94 | x | x |
| 814P | Papenburg, Ems | E | 01.01.1989-31.12.2015 | 27.0 | 98 | x | x |
| 816P | Herbrum, Hafendamm, Ems | M | 01.01.1989-01.11.2015 | 26.8 | 82 | x | |
| | | | | | Number of tide gauges | 111 | 98 |

*Competing interests.* The authors declare that they have no conflict of interest.

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
