# Peer review of "Reassessment of long-period constituents for tidal predictions along the German North Sea coast and its tidally influenced rivers"

_Ocean Science, 2019_

## Referee Comment (RC1) · Anonymous Referee #1 · 8 Jul 2019

For those readers interested in the analysis and prediction of tides, this paper is of considerable interest. The main advancement, and nominally the justification for publication, is a small revision to the tidal frequencies used by the German hydrographic office for their official predictions. But personally I found the paper of interest not for that, but rather just for the description of the HRoI method, about which I was completely unaware. Evidently developed by Horn in the 1950s, it is an unusual approach to tidal analysis. I'm especially appreciative of the fact it is developed for use in analyzing high and low water (rather than the more standard hourly data), and I can foresee

more applications of the method once readers become familiar with it. Indeed, there are old methods for analyzing high-low data by Schureman and Doodson, and probably others, but these are difficult to follow and probably no longer merit consideration with modern computers. The few papers on analysing high-low data that have been published in the last several decades – at least those I'm familiar with – are trivial and not worth reading. So I'm happy to have this new paper that explains Horn's methods.

The one drawback of the method, according to how the authors describe it, is on page 8, lines 10-15, where it seems a 19-year time series is needed. That is a pity. The authors find a way to overcome this limitation in the German network, since there are other stations around, but this is not always the case if the method is to be applied elsewhere.

For my own interest, I would liked to have seen more standard methods of prediction included in the tests of Tables 5-6, but I won't insist on this, because it would involve the authors using methods they may not have ready at hand. Others can perform this extended testing.

I have a few minor items to address; otherwise, I think OS should publish this paper.

The paper is well-written and the English is quite good, but there is a number of mis-spellings which I noticed. The authors should run an English-language spell-checker on the text to pick these up. But a spell-checker may not catch: page 5, line 17: frequency depended –> frequency-dependent page 4, line 78: what is "appodization" ?

Page 4, line 51: I understand why lunar transit times are computed, as they are fundamental to the method, but I do not understand why "lunar coordinates" are also needed. Or do the authors mean merely the mean longitudes needed to evaluate the Doodson arguments?

Page 4, lines 57-60: Regarding removal of "extreme events" – were these data also removed when the tests of Tables 5-6 were computed? Or do Tables 5-6 include ALL

data from 2016 ?

Do any of the German stations experience a double high tide? This occurs in some locations in the English Channel. If that occurs, how does the time indexing change?

---

## Referee Comment (RC2) · Anonymous Referee #2 · 25 Jul 2019

The concept underlying the paper involves a method to compute water level in a context of high tide water (HW) and low tide water (LW) prediction in a semi-diurnal regime. The presented study, going with developments and signal processing, is focused on computational method and an operational direction comes through quite clearly due to a need for quick computations and robust results. The paper is descriptive and the founding principle is based on two main existing technics with on the one hand a harmonic analysis (Doodson, etc.) and on the second hand, a non harmonic method used to take into account non linear waves interactions, partial tides and the non tidal dy-

namics. I think that HRoi method is a smart development for situation fitting the domain of validity introduced in the paper. Updated results along the German north sea coast show it's an effective method in this context; context well identified in the abstract. The presented processing and method offer an interesting view on tide prediction in coastal area where tide and river flux impact dynamics. My first feedback is that it is interesting to see the papers approach providing the method description and I appreciate the concise paper writing where the method constraints and objectives are clearly presented. In the paper, there is a part of discussions addressing the importance of observation pre-processing (ex: p8) and of computation conditioning (frequency analysis, noise reduction, regression, least squares minimization). It raises the question of nonlinear dynamics and the paper proposes a way to deal with it that fit the HRoi objective. The uncertainty for each step (observation, method, HW/LW prediction) is to my opinion an important part of the water level forecast and tide prediction. This contribution addresses part of the uncertainty question. I think it could be interesting to add few words to analyse the results presented in the paper, in terms of uncertainties or in standard deviations diagnostics, particularly in coastal areas with environmental challenges. This could echo the standard deviation values presented in the tables of the paper. I have some suggestions and several items to address. Otherwise, I think this paper describes a method and presents results on high and low water tidal prediction that are valuable to be published in Ocean Science, after having taken into account the remarks hereafter.

Please, see my suggestions, questions and notes below: An important point is mentioned in sentence "the HRoi combines the best from the harmonic and the nonharmonic method." (p 6). I think it could also be written in abstract section to emphasize it from the very start of the paper.

Tide gauges sections provide a good and updated view of the large tide gauges network. Reading the paper, the sensitivity of the results to the time series length is in our mind. This point is well discussed in the paper. I'm interested in knowing the time

sampling for tide gauges records.

Independently, did you use time series sampling (1) every x (hours? minutes?) or did you use (2) HW, LW recorded time series values? Following the method, keeping with the developments and discussion, it should be answer (1). Could you confirm?

"tidal events are irregularly spaced in time. Additionally, there are many longer data gaps which cannot be interpolated." (section 4.2, p 7). What is the maximum gap you observed in the tide gauges time series? And the longest continuous time series?

The last sentence, page 10, is important. The fact that this relates to parameters introduced in table 2, the fundamental variables, could be added in a note or in bracket.

Fig. 4 displays the periodogram of lunitidal intervals (L) after normalization. I understand that the normal variable (value) is the maximum value of lunitidal intervals. Fig. 4, I think it is useful to add in legend, the normalization variable in order to define what is the reference variable used for normalization. It's simple note but it drives the results and plots; Same suggestion for height variable.

I understand from ms, mh, mp, mN' fundamental parameters limits selection specific for this study (p 11) that these expressions are introduced in the text because they are useful for functions arguments development. But, this development is not included in the paper, nor cited/referenced. I'd say that if this part can't be used in the paper to help understanding the discussion, results or development, it could be removed from the text. But, if I'm wrong and if these expressions shouldn't be removed from text, then (1) addition of equations where these parameters are used would be useful for understanding or (2) one sentence could be added to say how it's useful to know the what type of selection have been done on ms, mh, mp, mN'.

P 12: Table 4: To get a short analysis of the percentage presented in this table, what is the importance of the main partial tide? Could you precise in table 4 legend, that the results of the most influencing tidal wave is a synthesis from all the selected tide

gauges?

P 17: Could you confirm that the results (fig. 7, p17) are residuals representative of both HW, LW? I think yes if I'm referring to fig 8. and 9, later in the paper. Fig. 7 validates the method in the frame of HW, LW prediction. Writing the residual mathematical formula is needed, I think, to sustain the text above fig. 7.

Just a suggestion: HW/LW prediction improvement percentage presented in tables could be completed by few words to provide some elements of analysis and understanding, to follow the reassessment.

For my interest, I'd like to see a result based on harmonic analysis and least squares minimization for the region of interest, in order to be able to compare its capacity to solve tidal dynamics to the HRoi method presented in the paper (for example in section 5). But therefore, I understand that the authors would have to make some other computations using tools and different methods from those which are presented and used here. So it's more a point for future discussion. Is the paper the first publishing for HRoi of investigations on long period constituents, as it is written in the paper? I'm not aware about the previous HRoi investigations publications for long time period constituents.

Legends and notations: Please see my suggestions and notes: P 2: angular velocities: After the word in text "omega", the mathematics notation $\omega i$ could be introduced, because it's used later, as the first reference in the text.

P 3: - y ˆ to be define in legend (equation 1) (predicted value I think, with y for height or lunitidal interval ). I think adding units in equation 1 is needed. Eq.2: symbol L for partial tide: Doodson reference for Eq.1 and Eq. 2 should be cited. They are derived from Laplace and Doodson theory and from harmonic analysis technics. Particularly, Doodson number is in the first column in tables 2 and 3 and is more generally a number currently used in tidal studies.

P 4: In table 2, I think there is a need to write the thinking who makes you remove the

fundamental parameter tau ($\tau$), respective to first letter in Doodson notation? $\tau$, s, h, p, N' for fundamental parameters to describe tides.$\tau$ to refer to hour angle of "mean" Moon. Just a suggestion: If you think it's relevant, I'd move table 2 in annex for it to play its role of quantitative reference (Tab 2 section 2).

P 5: Could you give a clear distinction between tn symbol used (p 4, p 5) and nt = lunar transit number (p3)? Reading page 5 and section 4.2 (page 7), could you write tn versus nt? It'd ease the reading and ease the comparison withp3, when transit number is introduced. Its formal symbol should be written (first sentence below table 1).

P 7: section 4.1 Data preparation: For both, Âń lunar transit number", Âńthe calculation of lunitidal intervals" my opinion is that adding symbols would be benefit for reading. nt and y ˆ (I suggest).

P 10: May I ask you to add slight modification to Li, Lh expression adding legend and adding units in these 2 expressions. I think it could be good to read the units ex: of angular velocity degree per h (cf table 2)? Degree per tn and L units.

P 12: Table 4 could be inserted in table 3, by adding: column R (table 4) after column Nh [%] in table 3, column description/name (table 4) in table 3

Fig. 6 (p 16): I appreciate the synthetic view of figure 6. Suggestion: could you add if possible, the explanation of number above the figure (relating the upper points of partial tides [rank]).

If possible, it could be interesting to see on map fig.1, the location of Borkum tide gauge, Cuxhaven, Steubenhöft and Emden, Große Seeschleuse tide gauges used in the paper to highlight results.
* * *

---

## Referee Comment (RC3) · Anonymous Referee #3 · 3 Aug 2019

General comments

This work deals with reassessment and its processes of long-period constituents for improving tidal prediction accuracy from a conventional tidal analysis and prediction method, the Harmonic Representation of Inequalities (HRoI). I think that it is worth publishing in Ocean Science in terms of preserving a conventional tidal method after revising some weakness.

They asserted the slight improvement using the new set of constituents through just

one year (2016) verification. I recommend that the authors should conduct additional two year (2017-2018) comparison between prediction and observation to clearly show the improvement.

As the authors mentioned, the HRoI is not widely used in comparison with a 'standard harmonic analysis and prediction (HAP)' method even if it has the better computational efficiency. Is it because that the HRoI is not open to the public or inconvenient to use? Additionally, I wonder if tidal prediction accuracy for the HRoI is better than that of the HAP. Can it predict tides at any time interval like the HAP? I think that the authors need to explain the additional reason why the HRoI is still used at BSH but most of countries have not used it. What are the advantages of using this method?

Some specific comments follow to help the authors address their manuscript's weakness: 1. On p. 2 line 4: 44 angular velocities -> 45 angular velocities (Need to check it)

2. In Table 3 and Table 4, angular velocity ($\omega$) should be expressed more than seven decimal places.

3. On p. 6 line 21: The authors need to explain how to determine the criteria of 60% of high and low waters in more details. It seems to me that the value is low. As shown in Table A1, there are a lot of data sets with more than 90% completeness.

4. On p. 7 line 12: What is 'tidal events'?

5. On p. 19 lines 1 and 2: in the residua -> in the residual (?); the two residua -> the two residual (?)

6. In Figure 7: The authors need to explain how to determine a bin width for time and height differences.

7. The authors need to use the subscript in expressing name of tidal constituents throughout the manuscript. That is, Sa -> $S_a$ (subscript a)

---

## Editor Comment (EC1) · Philip Woodworth (Editor) · 18 Aug 2019

18 August 2019

Editor comments on "Reassessment of long-period constituents for tidal predictions along the German North Sea coast and its tidally influenced rivers" by Andreas Boesch and Sylvin Müller-Navarra

I have looked at the 3 reviews and your replies - many thanks for those.

[Figure]

I agree with all 3 reviewers that it would be interesting to see this paper published, and as R1 says it would be useful in making the HRoI method more widely known. I have a few small comments myself below. And then I look forward to seeing a revised version.

One is that paper does have the feel of a highly-technical internal report and it might help to have an introductory paragraph in Section 6 (perhaps) to show that you know that there have been other methods for analysing HL waters in the past. Doodson (1951, IHO Special Publication No. 36) is the obvious one, but there is also a short discussion of the history in Bruce Parker's NOAA Tidal Analysis and Predictions Manual (NOS CO-OPS No. 3, p106-109) which you might wish to refer to.

Another is the comment by R1 about comparing the method used here to more standard harmonic methods, which you replied to in your paragraph (3) saying this was work in progress. But surely a tidal agency like the BSH is called on to produce hourly (or similar) tidal values for use in science or practical applications and you must have those data sets to hand. As regards the present paper it would not take much work to make a comparison for one or two places (say Cuxhaven). Last year I picked up a leaflet at the BSH which says'complete predictions of water level curves at Cuxhaven have been available on the internet since May 2010'.

Finally, on pages 6-8 or so I got a little lost with the discussion of the rankings. I understand the method for a particular station of course, but the rankings must be different for different stations so I was unclear how you arrived at the final choice. Could you make that clearer?

I also agree with R1 that, while the paper is completely understandable, the text could stand looking over by a native English speaker. I list a few trivial suggestions below and some other odd remarks.

p1, 15 long time series data

36 has been used

[p2, 11 - as an aside, I wondered at this point what Lubbock would have made of this paper. It is nice to see a method which has a little more physics in it than the harmonic method.]

60 eight time series

89 Could you give a reference to the Alphabetical Doodson Number e.g. the IHO Harmonic Constants Specification 2006? As far as I know this is not used much by the ocean tide community (it was invented by the UKHO I think) although does no harm to include it of course.

p4, 12 - on the German coast and in rivers

23 at the BSH tidal

26 tide gauges at Cuxhaven

40 The locations of all tide gauges

Figure 1 caption: Locations of tide gauges in the German Bight from Table A1.

53-60 I am not sure about this. I suspect that when most agencies produce tidal constants for a particular year they do not remove big storms; they are part of the sea level climatology, leading inevitably to ambiguity as to what defines the tide. So, in your case does this storm surge removal make any difference to the results?

68 as a spectral

Figure 2 caption line 3: Notice the upper part of the logarithmic scale is truncated

p5, 10 - time series

17 frequency-dependent

p7, 21 This is demonstrated in the lower panels

42 Conversely, the partial tide

45 I would add '(ms <= 8)' after 'month'

p8, 7 which occur at R=4 in the list cannot

17 - you might also like to refer to papers by Amin which present departures from nodal variation from the equilibrium tide expectations in this region.

p9, 12 fulfil

Figure caption 7, line 2: add (p.t.) after tides so the insert is explained

Tables 5 and 6 - is it necessary to have gauge number in these tables

32 percentage changes

38 residuals

39 for the two sets of residuals (times and heights)

45 no major improvements

49 peaks are clearly reduced.

Figure 8-11. It might be good to make 8 and 10 into 8(a,b) and 9 and 11 into a new 9(a,b). Define (p.t.) in the caption again.

p10, 29 four frequencies were reduced

p11, 3 will be used

p13, 4 correspond to

---

## Author Response (AR1)

**Final author response: "Reassessment of long-period constituents for tidal predictions along the German North Sea coast and its tidally influenced rivers"**

We would like to thank again the three reviewers and the journal editor for their constructive feedback. Detailed point-to-point responses to all referee comments can be found in the respective author comments from the discussion phase (repeated below for completeness).

The changes to the manuscript are explained in the individual answers in the author comments. One major change (and a deviation from the answers to the reviewers) is the inclusion of a new section 6 which provides a comparison of predictions made with the Harmonic Representation of Inequalities and with the Harmonic Method for the two tide gauge locations Cuxhaven and Hamburg. We hope that this analysis satisfies the repeatedly expressed wish for a comparison of the two prediction methods. Additionally, some minor editorial corrections have been applied (e.g. corrections for spelling or sentence structure).

Please find the marked-up manuscript version showing the changes made below the author comments.

Best regards,
Andreas Boesch and Sylvin Müller-Navarra

**Authors' response to Referee Comment #1**

Dear Reviewer #1.

Thank you very much for your attention to the manuscript and your review report. Please find below our replies to your comments. The different items from the review report are first cited, followed by our responses.

1) "But personally I found the paper of interest not for that, but rather just for the description of the HRoI method, about which I was completely unaware. Evidently developed by Horn in the 1950s, it is an unusual approach to tidal analysis. I'm especially appreciative of the fact it is developed for use in analyzing high and low water (rather than the more standard hourly data), and I can foresee more applications of the method once readers become familiar with it."

>> Yes, this paper is also an opportunity to (re-)introduce the Harmonic Representation of Inequalities (HRoI) to the scientific community. Although the method has been around for a long time, many tidal scientists are not aware of it (anymore). We agree that the method can be interesting for others, especially when studying tides in estuaries or when tide gauges run dry around low water.

2) "The one drawback of the method, according to how the authors describe it, is on page 8, lines 10-15, where it seems a 19-year time series is needed."

>> Best results are, of course, achieved when using a 19-year time series, because in this case all relevant tidal information is contained within the data. However, shorter time series can also be analysed. The rank R in table 4 indicates which partial tides need to be dropped in the case of shorter time series. In operational usage, we use the method directly if 10 years or more years are available. For time series between 10 and 19 years, 5 of the 39 partial tides are dropped (see figure 6). Analyses with even shorter time series are also possible but with decreasing accuracy; in these

cases the transfer of a good prediction from a nearby station often gives better results.

3) "For my own interest, I would liked to have seen more standard methods of prediction included in the tests of Tables 5-6, but I won't insist on this, because it would involve the authors using methods they may not have ready at hand. Others can perform this extended testing."

>> The comparison of the HRoI with other methods (e.g. the harmonic method) is not the subject of this paper and would be beyond its scope. We agree that this testing is interesting and important. We are starting to develop tools for extensive comparison of the HRoI and the harmonic method, and we will share these results with the scientific community when results are available in the future. We also invite others to use the HRoI for their applications and comparisons.

4) "The paper is well-written and the English is quite good, but there is a number of misspellings which I noticed. The authors should run an English-language spell-checker on the text to pick these up. But a spell-checker may not catch: page 5, line 17: frequency depended –> frequency-dependent page 4, line 78: what is "appodization" ?"

>> We will look carefully through the manuscript to catch the remaining spelling mistakes. The discussion paper version (one column) had already been improved in this regard, compared to the initially uploaded two-column version (which you probably read according to the cited pages and line numbers).
The word "appodization" should be spelled "apodization" and is a window function that is applied (multiplied) to the data in order to reduce side lobes in the periodogram. Otherwise, these side lobes could be identified as true signals by mistake. In the revised manuscript, we will add more information and references in the corresponding paragraph.

5) "Page 4, line 51: I understand why lunar transit times are computed, as they are fundamental to the method, but I do not understand why "lunar coordinates" are also needed. Or do the authors mean merely the mean longitudes needed to evaluate the Doodson arguments?"

>> The lunar transit times are computed using the algorithm published in chapter 15 of Meeus (1998). Inputs to this algorithm are the right ascension and declination of the transiting body, i.e. the moon. These coordinates are calculated using the lunar theory by Chapront-Touzé and Chapront (1991). We will make the corresponding sentence clearer in the revised manuscript.

6) "Page 4, lines 57-60: Regarding removal of "extreme events" – were these data also removed when the tests of Tables 5-6 were computed? Or do Tables 5-6 include ALL data from 2016 ?"

>> The observed water levels of the year 2016 used for comparison are also filtered as described in Sect. 4.1. This way, the tidal predictions are compared to observations that represent the tidal behaviour better than the full data sets including extreme events. We will add this information explicitly in Sect. 5.1.

7) Do any of the German stations experience a double high tide? This occurs in some locations in the English Channel. If that occurs, how does the time indexing change?

>> There are no German stations with a double high tide. In its current form, the Harmonic Representation of Inequalities is tailored to strictly semi-diurnal tides because it was developed for the conditions in the German Bight. The possible

adaption to other tidal forms is a very interesting question. If the double high tide appears in every cycle (and if this is known to the analyst), it should be no problem to introduce four more equations of the type of Eq. (1); one model equation for each of the heights and times of the second high water assigned to the upper or lower lunar transit. For arbitrary mixed types, the direct calculation of high and low waters with the HRoI seems not to be possible. In these cases, one needs to first calculate the full curve, either with the harmonic method or maybe with the extended HRoI (as mentioned in Sect. 2), and derive the minima/maxima from the curve.

**Authors' response to Referee Comment #2**

Dear Reviewer #2.

Thank you very much for your detailed review report. We appreciate that you have taken the time to read the manuscript and to comment on it. Please find below our replies to your comments. The different items from the review report are first cited, followed by our responses.

1) "An important point is mentioned in sentence "the HRoi combines the best from the harmonic and the nonharmonic method." (p 6). I think it could also be written in abstract section to emphasize it from the very start of the paper.

>> The quoted passage is taken from Horn(1960) and might be too subjective to be included in an abstract. However, it is a good idea to specify the nature of the HRoI in the abstract and we will add this information in the revised manuscript.

2) "I'm interested in knowing the time sampling for tide gauges records.
Independently, did you use time series sampling (1) every x (hours? minutes?) or did you use (2) HW, LW recorded time series values? Following the method, keeping with the developments and discussion, it should be answer (1). Could you confirm?"

>> The recorded sampling rate of the tide gauges is 1 minute (for about the last 20 years, depending on the individual tide gauge; previously only high and low water data were available). The presented prediction method uses time series of times and heights of high and low water (and predicts only times and heights of high and low water).

3) "What is the maximum gap you observed in the tide gauges time series? And the longest continuous time series?"

>> The longest continuous time series are from Cuxhaven and Hamburg, each with 115 years and a maximum data gap of half a day, which means that no more than one high or low water is missing at a time. The maximum gaps can be longer than 10 years but the 60% criterion ensures that a sufficient amount of data is available from each tide gauge.

4) "The last sentence, page 10, is important. The fact that this relates to parameters introduced in table 2, the fundamental variables, could be added in a note or in bracket."

>> Thank you for this remark. We will add a note that the parameters, for which we define the ranges on page 11, are related to the parameters in Table 2. Furthermore, we will rework this part of Sect. 4.3 to improve the readability (see also item 5, 6, 16).

5) "Fig. 4 displays the periodogram of lunitidal intervals (L) after normalization. I understand that the normal variable (value) is the maximum value of lunitidal intervals. Fig. 4, I think it is

useful to add in legend, the normalization variable in order to define what is the reference variable used for normalization. It's simple note but it drives the results and plots; Same suggestion for height variable."

>> see answer to comment 16.

6) "I understand from ms, mh, mp, mN' fundamental parameters limits selection specific for this study (p 11) that these expressions are introduced in the text because they are useful for functions arguments development. But, this development is not included in the paper, nor cited/referenced. I'd say that if this part can't be used in the paper to help understanding the discussion, results or development, it could be removed from the text. But, if I'm wrong and if these expressions shouldn't be removed from text, then (1) addition of equations where these parameters are used would be useful for understanding or (2) one sentence could be added to say how it's useful to know the what type of selection have been done on ms, mh, mp, mN'."

>> We will rework this part of Sect. 4.3 (see also item 4, 5, 16) and include information on the calculation of the angular velocities (and add references to Section 2 and Table 2). The definition of the ranges of the linear coefficients m should stay in the manuscript as it sets the limits for the assignment of possible partial tides.

7) P 12: Table 4: To get a short analysis of the percentage presented in this table, what is the importance of the main partial tide? Could you precise in table 4 legend, that the results of the most influencing tidal wave is a synthesis from all the selected tide gauges?

>> The importance of the main partial tide (half synodic month) can be derived from table 3, where the average line intensities from the periodograms are listed. A quantitative statement (in physical units) is difficult, because of the averaging of data from different tide gauges and the normalization of the generalized Lomb Scargle periodogram.
We will make it clear in the legend of table 4 (and table 3) that the results are a synthesis from all the selected tide gauges

8) "P 17: Could you confirm that the results (fig. 7, p17) are residuals representative of both HW, LW? I think yes if I'm referring to fig 8. and 9, later in the paper. Fig. 7 validates the method in the frame of HW, LW prediction. Writing the residual mathematical formula is needed, I think, to sustain the text above fig. 7."

>> Yes, Fig. 7 shows the results for both high water and low water. Residuals are the difference between observed and predicted vertices (high or low water) with the same transit number and event index k. We will add the information on how the residuals are calculated in the revised manuscript.

9) "HW/LW prediction improvement percentage presented in tables could be completed by few words to provide some elements of analysis and understanding, to follow the reassessment."

>> We will add the formula for calculation of the changes and a few sentences about the contents (especially the extreme bins) of Fig. 8 and 9.

10) "For my interest, I'd like to see a result based on harmonic analysis and least squares minimization for the region of interest, in order to be able to compare its capacity to solve tidal dynamics to the HRoi method presented in the paper (for example in section 5). But therefore, I understand that the authors would have to make some other computations using tools and different methods from those which are presented and used here. So it's more a point for future discussion. Is the paper the first publishing for HRoi of investigations on long

period constituents, as it is written in the paper? I'm not aware about the previous HRoi investigations publications for long time period constituents."

>> This is the first publication in which the investigation of long term constituents for the HRoI is described. Older publications on the HRoI, as cited in the manuscript, only present the list of constituents without details about its preparation. We will add a sentence in the conclusions about the need to conduct a comparison study with the harmonic analysis to gain more insight on the relative performances.

11) "P 2: angular velocities: After the word in text "omega", the mathematics notation !i could be introduced, because it's used later, as the first reference in the text."

>> We will add the mathematics notation "omega" after "angular velocities" on page 2.

12) "P 3: - ŷ to be define in legend (equation 1) (predicted value I think, with y for height or lunitidal interval ). I think adding units in equation 1 is needed. Eq.2: symbol L for partial tide: Doodson reference for Eq.1 and Eq. 2 should be cited. They are derived from Laplace and Doodson theory and from harmonic analysis technics. Particularly, Doodson number is in the first column in tables 2 and 3 and is more generally a number currently used in tidal studies."

>> We add some information on variable names and units to better guide the reader and include the references to the alphanumeric Doodson number. The symbol J (not L) is correct in Eq. 2 as the sum runs over the J data points.

13) "P 4: In table 2, I think there is a need to write the thinking who makes you remove the fundamental parameter tau ( ), respective to first letter in Doodson notation? , s, h, p, N' for fundamental parameters to describe tides. to refer to hour angle of "mean" Moon. Just a suggestion: If you think it's relevant, I'd move table 2 in annex for it to play its role of quantitative reference (Tab 2 section 2)."

>> By construction of the method (HRoI) only long-period constituents need to be used, i.e. the parameter belonging to tau is always equal to zero. We will revise the corresponding sentence. We think that Table 2 is more than a quantitative reference, but a central element of the paper that connects our work with previous studies and presents fundamental information for all constituents. Therefore, we would like to keep the table in the main part of the paper instead of moving it to the appendix.

14) "P 5: Could you give a clear distinction between tn symbol used (p 4, p 5) and nt =lunar transit number (p3)? Reading page 5 and section 4.2 (page 7), could you write tn versus nt? It'd ease the reading and ease the comparison withp3, when transit number is introduced. Its formal symbol should be written (first sentence below table 1)."

>> The symbol "tn" is the unit symbol for transit number (such as "h" for hour). The symbol "$n_t$" is the variable that stands for a value of the transit number (such as "t" for time). We move the introduction of the unit symbol to the sentence with the definition of the transit number. This gives a better distinction between the two notations.

15) "P 7: section 4.1 Data preparation: For both, ´n lunar transit number", ´nthe calculation of lunitidal intervals" my opinion is that adding symbols would be benefit for reading. nt and ŷ (I suggest)."

>> The symbol for the transit number is added in the corresponding sentence. The lunitidal intervals do not have a unique symbol (the symbol "y" can stand for lunitidal intervals or heights) and is therefore not included in the sentence. Instead, we included the symbol "y" in the first sentence in Sect. 4.2.

16) "P 10: May I ask you to add slight modification to Li, Lh expression adding legend and adding units in these 2 expressions. I think it could be good to read the units ex: of angular velocity degree per h (cf table 2)? Degree per tn and L units."

>> The expressions Li and Lh are unitless as the generalized Lomb-Scargle periodogram is normalized to unity. A value of 0 indicates no improvement of the fit and a value of 1 a "perfect" fit (see Zechmeister and Kürster 2009, full reference in the manuscript). We will add this information on page 10.

17) "P 12: Table 4 could be inserted in table 3, by adding: column R (table 4) after column Nh [%] in table 3, column description/name (table 4) in table 3"

>> We think that table 3 and table 4 should be kept separate. Table 3 shows the results from the initial analysis based on the defined rules. Table 4 shows the final results, after manual adjustments have been made to the selection of partial tides. Keeping these two tables (which also belong to different sections) separate makes the procedure more transparent.

18) "Fig. 6 (p 16): I appreciate the synthetic view of figure 6. Suggestion: could you add if possible, the explanation of number above the figure (relating the upper points of partial tides [rank])."

>> The numbers at the top of the figure are just the counts of partial tides (number of circles) in each "column". This is mentioned in the caption.

19) "If possible, it could be interesting to see on map fig.1, the location of Borkum tide gauge, Cuxhaven, Steubenhöft and Emden, Große Seeschleuse tide gauges used in the paper to highlight results"

>> Thanks for this great suggestions. These three stations (and Hamburg) will be highlighted in Fig.1 in the revised manuscript.

**Authors' response to Referee Comment #3**

Dear Reviewer #3.

Thank you very much for your time and effort to review our manuscript. Please find below our replies to your comments. The different items from the review report are first cited, followed by our responses.

1) "They asserted the slight improvement using the new set of constituents through just one year (2016) verification. I recommend that the authors should conduct additional two year (2017-2018) comparison between prediction and observation to clearly show the improvement."

>> The main focus of the manuscript is the preparation of the list of tidal constituents as used by the operational tidal forecasting service. We think that in this context the verification over one year is sufficient. The old sets of constituents have proven over several decades to deliver good results and no major differences were expected. The presented comparison shows that the new set of constituents can be expected to work equally well or even better because several frequencies in the residuals are now removed (see Fig. 10 and 11). We argue that an additional comparison over two years is not likely to show any significantly different results, but rather inflate the manuscript unnecessarily. Furthermore, quality-checked times and heights of high

and low waters are not yet available from the respective authorities for several tide gauges for the year 2018. This would limit the comparability of the different verifications. We hope that the referee understands our arguments and does not insist on further verification studies.

2) "As the authors mentioned, the HRoI is not widely used in comparison with a 'standard harmonic analysis and prediction (HAP)' method even if it has the better computational efficiency. Is it because that the HRoI is not open to the public or inconvenient to use? Additionally, I wonder if tidal prediction accuracy for the HRoI is better than that of the HAP. Can it predict tides at any time interval like the HAP? I think that the authors need to explain the additional reason why the HRoI is still used at BSH but most of countries have not used it. What are the advantages of using this method?"

>> We are not in a position to judge why the HRoI is not used more widely. The method has been published (as referenced in the manuscript) and it is fairly easy to use.
The original implementation of the HRoI, as described in Sect. 2, uses recorded time series of times and heights of high and low waters in order to predict times and heights of high and low waters. The concept of the HRoI can be generalized to determine the full tidal curve based on equally spaced water level records (e.g. 10 minute intervals). This generalized concept is explained in Müller-Navarra (2013; full reference in the manuscript) and is not subject of the analysis presented in the manuscript.
The characteristics (including advantages) of the HRoI are mentioned in Sect. 1 and 2, but we agree that this aspect is scattered throughout the two sections and should be cleaned up and expanded. In the revised manuscript, we will remove the last paragraph of Sect. 2 and insert it after the second paragraph of Sect. 2. The paragraph will also be expanded to address the advantages in more detail.
The comparison of the HRoI with other methods (e.g. the harmonic method) is not the subject of this paper and would be beyond its scope. Reliable harmonic constants exist only for a few German tide gauges and a comparison study of this kind would need a lot of resources that are not available at present. We agree that this testing is interesting and invite others to use the HRoI for their applications and comparisons.

3) "On p. 2 line 4: 44 angular velocities -> 45 angular velocities (Need to check it)"

>> The list of partial tides published by Horn (1960) consists of 44 angular velocities. These 44 angular velocities are also marked in our Table 2. The sentence in the manuscript is correct.

4) "In Table 3 and Table 4, angular velocity (!) should be expressed more than seven decimal places."

>> We will add one decimal place to the angular velocities in Tables 3 and 4 in the revised manuscript; also to make it consistent with the angular velocities in Table 2 and the operational usage. More decimal places would be beyond the uncertainty estimate which is of the order of 1e-7 degrees/transit number.

5) "On p. 6 line 21: The authors need to explain how to determine the criteria of 60% of high and low waters in more details. It seems to me that the value is low. As shown in Table A1, there are a lot of data sets with more than 90% completeness."

>> This selection criterion ensures that only data from tide gauges that record both high and low water are included in the analysis. Some tide gauges fall dry at low water and do not record meaningful tidal data during this time (and no low water height and time is included in the quality-checked time series). These tide gauges

have a data completeness of 50% at most. The threshold at 60% is rather conservative in this regard.

6) "On p. 7 line 12: What is 'tidal events'?"

>> High water and low water are referred to as "tidal events". We will clarify the language in the revised manuscript.

7) "On p. 19 lines 1 and 2: in the residua -> in the residual (?); the two residua -> the two residual (?)"

>> Yes, this is a typo. The sentence should read "The change of constituents has an influence on the remaining periodicities in the residuals." This will be corrected in the revised manuscript.

8) "In Figure 7: The authors need to explain how to determine a bin width for time and height differences."

>> The number and the width of the bins are chosen in such a way that the central bin is centred at the origin.

9) "The authors need to use the subscript in expressing name of tidal constituents throughout the manuscript. That is, Sa -> S_a (subscript a)"

>> We followed the naming scheme of the "Standard list of Tidal Constituents" by the IHO which does not use subscripts. We will add this information on page 6, line 2 in the revised manuscript, but prefer to keep the naming as it is if the reviewer does not have any objections.

**Authors' response to Editor Comment #1**

Dear Phil,

Thank you very much for your additional comments to our manuscript. Please find below our replies to your comments. We will upload the revised manuscript as soon as possible.

1) "One is that paper does have the feel of a highly-technical internal report and it might help to have an introductory paragraph in Section 6 (perhaps) to show that you know that there have been other methods for analysing HL waters in the past."

>> We will add some more references to other methods of tidal predictions in the revised manuscript. Thank you for your literature suggestions.

2) "Another is the comment by R1 about comparing the method used here to more standard harmonic methods, which you replied to in your paragraph (3) saying this was work in progress. But surely a tidal agency like the BSH is called on to produce hourly (or similar) tidal values for use in science or practical applications and you must have those data sets to hand. As regards the present paper it would not take much work to make a comparison for one or two places (say Cuxhaven). Last year I picked up a leaflet at the BSH which says'complete predictions of water level curves at Cuxhaven have been available on the internet since May 2010'."

>> The tidal information service from BSH does not provide hourly predictions on an operational basis (yet). As we start to have 19 years of 1-minute data from more and

more tide gauges, we are currently setting up the programs to use this high resolution data in our routine tidal predictions. For previous years, only the HL water recordings were saved for most tide gauges. As the request for a comparison with the more widely used harmonic methods has been expressed in all review reports, we will include a short comparison for two stations (probably Cuxhaven and Hamburg) in the revised manuscript.

The leaflet that you are referring to probably covers the water level and storm surge forecasting service (not the tidal information service). These water level curves (with and without surge) are produced by different methods, in which the tidal data calculated with the HRoI (times and heights of high and low waters) is used as an input.

3) "I understand the method for a particular station of course, but the rankings must be different for different stations so I was unclear how you arrived at the final choice. Could you make that clearer?"

>> The rankings displayed in Table 4 are a synthesis from the data of all analysed tide gauges. This will be mentioned in the caption of the table and will be made clearer in the corresponding paragraph. The goal of Table 4 is to produce one comprehensive list that reflects the information from all tide gauges in the area under investigation. A tailored analysis for an individual tide gauge is of course possible (and needed), if the general list does not lead to good results.

4) Most of the other remarks have been directly incorporated into the revised manuscript (thanks for all the details). Here are the answers to your questions:

4a) "I suspect that when most agencies produce tidal constants for a particular year they do not remove big storms; they are part of the sea level climatology, leading inevitably to ambiguity as to what defines the tide. So, in your case does this storm surge removal make any difference to the results?"

>> We try to predict water levels considering past long-term meteorological conditions (as good as this is possible). A single extreme event, like a severe storm surge, is not representative of the tidal behaviour at a site (and cannot be forecasted in the framework of tidal predictions). The model function (sum of harmonics) is also not made to properly account for these extreme events and the least squares method is likely (depends on the number of storm surges) to give results that lead to slightly higher heights at all times, if the storms are nor removed. We do not have numbers at hand on how much this storm surge removal influences the results (this will also depend on the number of extreme events in an individual time series). Part of this topic is the fundamental question on how to define the (astronomical) tide.

4b) "Tables 5 and 6 - is it necessary to have gauge number in these tables"

>> The numbers are necessary, because the short names for the tide gauges are not always unique, e.g. Borkum (Fischerbalje) vs. Borkum (Südstrand).

4c) "Figure 8-11. It might be good to make 8 and 10 into 8(a,b) and 9 and 11 into a new 9(a,b)."

>> We would like to keep these figures separated, as they cover slightly different aspects of the residual analysis.

[revised manuscript text omitted]